# DNA variants affecting the expression of numerous genes in *trans* have diverse mechanisms of action and evolutionary histories

**Sheila Lutz**, **Christian Brion**, **Margaret Kliebhan**, **Frank W. Albert** *

Department of Genetics, Cell Biology, & Development, University of Minnesota, Minneapolis, MN, United States of America

* falbert@umn.edu

**Data Availability Statement:** Analysis code and raw plate reader data are provided at https://github.com/frankwalbert/threeHotspots. All RNA Sequencing data are available from the GEO

## Abstract

DNA variants that alter gene expression contribute to variation in many phenotypic traits. In particular, *trans*-acting variants, which are often located on different chromosomes from the genes they affect, are an important source of heritable gene expression variation. However, our knowledge about the identity and mechanism of causal *trans*-acting variants remains limited. Here, we developed a fine-mapping strategy called CRISPR-Swap and dissected three expression quantitative trait locus (eQTL) hotspots known to alter the expression of numerous genes in *trans* in the yeast *Saccharomyces cerevisiae*. Causal variants were identified by engineering recombinant alleles and quantifying the effects of these alleles on the expression of a green fluorescent protein-tagged gene affected by the given locus in *trans*. We validated the effect of each variant on the expression of multiple genes by RNA-sequencing. The three variants differed in their molecular mechanism, the type of genes they reside in, and their distribution in natural populations. While a missense leucine-to-serine variant at position 63 in the transcription factor Oaf1 (L63S) was almost exclusively present in the reference laboratory strain, the two other variants were frequent among *S. cerevisiae* isolates. A causal missense variant in the glucose receptor Rgt2 (V539I) occurred at a poorly conserved amino acid residue and its effect was strongly dependent on the concentration of glucose in the culture medium. A noncoding variant in the conserved fatty acid regulated (FAR) element of the *OLE1* promoter influenced the expression of the fatty acid desaturase Ole1 in *cis* and, by modulating the level of this essential enzyme, other genes in *trans*. The *OAF1* and *OLE1* variants showed a non-additive genetic interaction, and affected cellular lipid metabolism. These results demonstrate that the molecular basis of *trans*-regulatory variation is diverse, highlighting the challenges in predicting which natural genetic variants affect gene expression.

database as series GSE134169. All remaining data are within the manuscript and its Supporting Information files. Plasmids are available at Addgene under IDs 131774 and 131784.

**Funding:** This work was supported by National Institutes of Health (NIH; https://www.nih.gov) grant R35GM124676 to FWA. The funders had no role in study design, data collection and analysis, decision to publish, or preparation of the manuscript.

**Competing interests:** The authors have declared that no competing interests exist.

## Author summary

Differences in the DNA sequence of individual genomes contribute to differences in many traits, such as appearance, physiology, and the risk for common diseases. An important group of these DNA variants influences how individual genes across the genome are turned on or off. In this paper, we describe a strategy for identifying such "*trans*-acting" variants in different strains of baker's yeast. We used this strategy to reveal three single DNA base changes that each influences the expression of dozens of genes. These three DNA variants were very different from each other. Two of them changed the protein sequence, one in a transcription factor and the other in a sugar sensor. The third changed the expression of an enzyme, a change that in turn caused other genes to alter their expression. One variant existed in only a few yeast isolates, while the other two existed in many isolates collected from around the world. This diversity of DNA variants that influence the expression of many other genes illustrates how difficult it is to predict which DNA variants in an individual's genome will have effects on the organism.

## Introduction

DNA variants that alter gene expression are an important source of genetic variation for many traits [1], including common disease in humans [2], agricultural yield [3,4] and evolutionary change [5]. To map gene expression variation in the genome, expression levels are measured in a population of individuals and related to the genotype of each individual. This approach identifies expression quantitative trait loci (eQTLs)–genomic regions that each contain one or more variants that affect gene expression.

eQTLs can be classified into two types based on their mechanism of action. *Cis* eQTLs arise from variants that alter the expression of genes on the same DNA molecule, for example by changing the sequence of a regulatory element in a promoter. Most *cis*-acting variants are located close to or within the genes they influence, such that *cis* eQTLs can be detected as "local" eQTLs that overlap the given gene. By contrast, *trans* eQTLs arise from variants that change the activity and/or abundance of a diffusible factor which in turn alter the expression of other genes. *Trans* eQTLs can be located anywhere in the genome relative to the genes they affect. While they can be local (e.g., if a gene encoding a transcription factor resides next to a gene targeted by that factor), most *trans* eQTLs are "distant" from the genes they affect, usually on different chromosomes.

As in genetic mapping of other traits, identifying the specific DNA variants that have causal effects in eQTL regions is challenging. Recent studies have made progress in identifying *cis* acting variants (e.g. [6–10]). However, few *trans*-eQTLs have been resolved to single variants. This is although most of the heritable contribution to gene expression variation arises from *trans* rather than *cis* eQTLs [11–16], and although *trans* acting variation is likely to play pivotal roles in shaping diseases and phenotypes within species [11,17]. Identifying the molecular nature of *trans*-acting variants and the mechanisms by which they alter gene expression is key to understanding the connection between genotypic and phenotypic variation. In particular, knowledge of causal variants will allow us to examine their population frequency, distribution and plausible evolutionary histories, ask if and how the same variants influence phenotypes other than gene expression, and ultimately improve attempts to predict the consequences of genome sequence variation on the organism.

Natural isolates of the yeast *Saccharomyces cerevisiae* have provided fundamental insights into the genetics of gene expression variation ([12,18–34], reviewed in [35]). Particularly

intensive efforts have been directed at the comparison between a laboratory strain (the genome reference strain S288C, or "BY") and a wine strain (RM11a, "RM"), whose genomes differ at about 40,000 variants. eQTL mapping in recombinant progeny from a cross between these strains revealed the existence of eQTL hotspots that each influence the expression of numerous genes in *trans* [18]. Many of these hotspots also affect protein levels [22,36,37].

Recently, an analysis of mRNA levels in an expanded set of 1,012 BY/RM progeny provided a more comprehensive view of regulatory variation in this cross [12]. Specifically, the 5,643 genes that were expressed in standard growth conditions were affected by more than 36,000 eQTLs, which together accounted for the majority of genetic variation in mRNA levels in this cross. The vast majority of eQTLs (92%) acted in *trans*, and more than 90% of these *trans*-eQTLs were co-located at 102 hotspots (Fig 1A). Some hotspots affected the expression of thousands of genes. Their predominance makes these *trans*-eQTL hotspots a promising reservoir for understanding the molecular basis of *trans*-acting variation. Indeed, a small number of hotspots in this and other yeast crosses have been resolved to their causal genes and nucleotide variants [18,20,23,30,31,34,38–41]. However, progress towards a more complete view of hotspot variants has been hampered by the challenges of engineering and measuring the expression effects of candidate variants.

Here, we describe a strategy for the identification of causal eQTL hotspot variants that combines a genome engineering approach with high-throughput quantification of fluorescently tagged protein expression as a phenotypic readout. We used this strategy to identify causal variants underlying three *trans*-acting eQTL hotspots in the BY and RM strains: a common missense variant in the glucose receptor Rgt2, a rare missense variant in the oleate-activated transcription factor Oaf1, and a common variant in the promoter of *OLE1* that alters the expression of this essential fatty acid desaturase gene. We studied the effects of these variants in more detail and discovered that the effect of the *RGT2* variant is influenced by the environment, and that the *OAF1* and *OLE1* variants interact in a non-additive fashion and lead to changes in cellular lipids. These results show that variants underlying *trans*-acting hotspots are highly diverse. They can be common or rare in the population, different in their evolutionary conservation, and located in the coding or noncoding region of genes encoding functionally diverse proteins.

## Results

### The CRISPR-Swap strategy for engineering allelic series

To assist fine-mapping of hotspot intervals, we devised "CRISPR-Swap" (Fig 1B), a strategy for efficient allele exchange that combines advantages of "insert-then-replace" methods [42,43] with CRISPR/Cas9 engineering [44–46]. First, a given locus is replaced with a selectable marker cassette. Second, the strain is transformed with a plasmid that expresses Cas9 and a guide RNA (gRNA) that targets the cassette, along with a DNA repair template containing terminal homology to sequences flanking the cassette in the genome. We designed the "gCASS5a" gRNA to specifically target a sequence shared by several selectable marker cassettes used for gene deletions in the popular pFA6a series (e.g., KanMX6, natMX4 and hphMX4; [47–50]; S1 Fig) such that the same gRNA can be used to replace each of these cassettes. By inserting two different cassettes flanking a genomic region, this gRNA can be used to exchange both cassettes along with the intervening sequence. This "double-cut" CRISPR-Swap method enables allele exchange even when the region contains sequences essential for survival (Fig 1B). Additionally, we designed a gRNA, "gGFP", that specifically targets cassettes used for C-terminal tagging of open reading frames with GFP.

The gRNA/Cas9 complex is constitutively expressed from the CRISPR-Swap plasmid and will continue to cleave at the cassette(s) in the genome until a repair is made that abolishes the

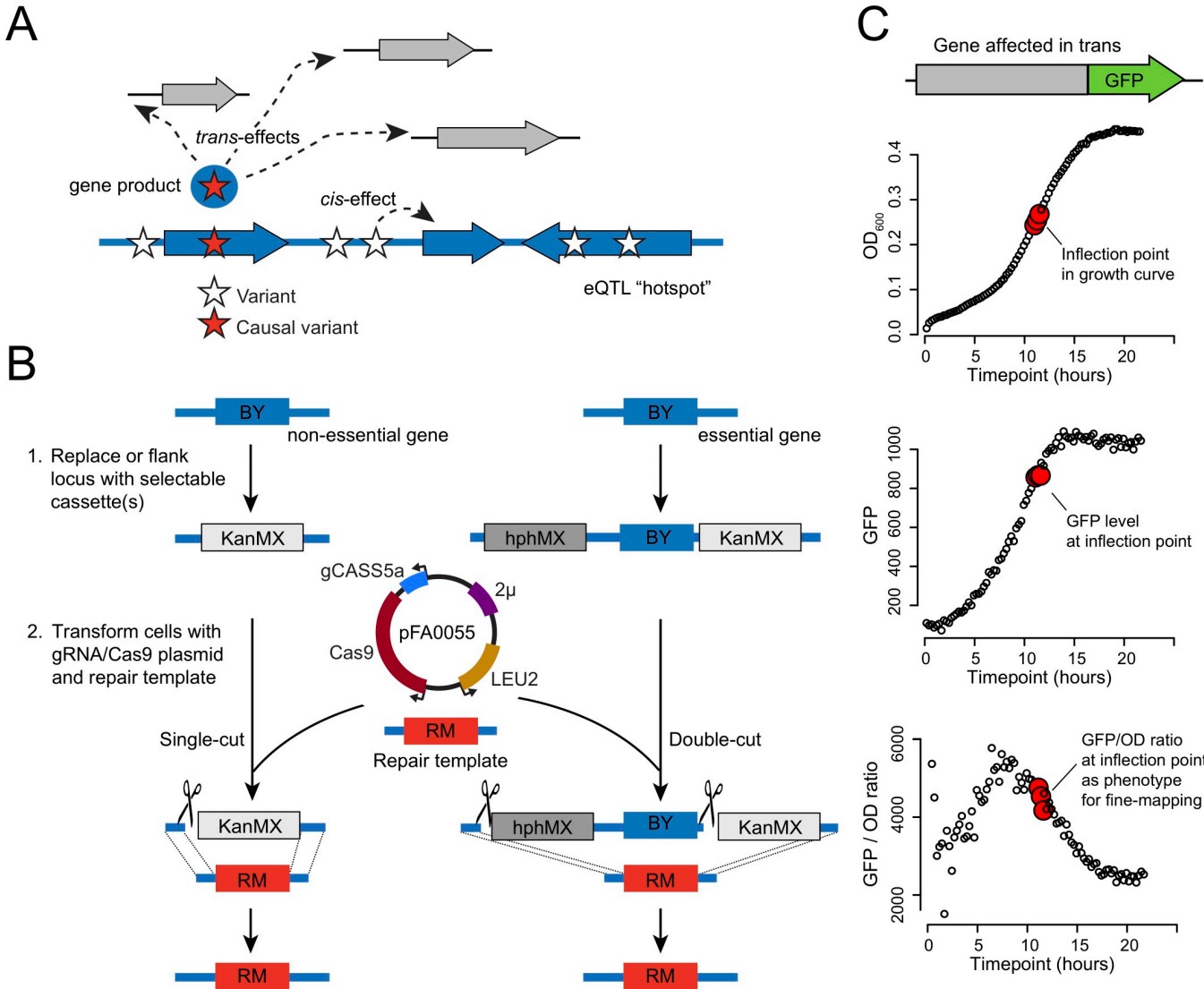

**Fig 1. Hotspot fine mapping strategy.** A. Illustration of the possible *cis* and *trans* effects at an eQTL hotspot. A genomic region, shown in blue with several genes depicted as wide arrows, harbors multiple variants (stars). Of these, one causal variant (red star) alters the activity and/or abundance of a gene product (blue circle with red star), which alters the expression of multiple genes (gray arrows) in *trans*. Another variant is shown affecting a neighboring gene in *cis*, but has no *trans* effect. B. A schematic showing two examples of engineering a BY allele (blue) to an RM allele (red) using CRISPR-Swap. First, a non-essential gene is replaced by the G418 resistance cassette (KanMX) or an essential gene is flanked by the hygromycin resistance (hphMX) and KanMX cassettes. Second, the strain is transformed with the CRISPR-Swap plasmid pFA0055 that expresses Cas9 and the guide RNA gCASS5a and the auxotrophic marker LEU2; and a PCR-generated repair template containing the desired RM allele. The gCASS5a/Cas9 complex directs cleavage (scissors) of the cassettes. Either a single-cut or a double-cut occurs depending on the number of cassettes present. Selection of transformants for leucine prototrophy and loss of G418, or hygromycin and G418, resistance identifies strains with the desired RM allele replacement. C. Quantifying the expression of a representative gene affected in *trans* by a hotspot. Fluorescence of the protein expressed from the GFP-tagged gene and optical density ($OD_{600}$) of the culture are measured in 15 minute intervals during growth in a plate reader. The inflection point, the point where the culture exits logarithmic growth, and two flanking points are used to determine the GFP/$OD_{600}$ ratio for phenotyping the effect of the engineered alleles.

gRNA recognition sequence or the cell dies. Consequently, after transformation, all colonies that form on media lacking leucine have undergone a repair that blocks further cleavage by the gRNA/Cas9 complex. We designed the gCASS5a and gGFP gRNAs to cleave in the cassette but outside of the selectable marker gene, such that marker expression should be maintained if repair occurs without exchange of the selectable marker (S1 Fig). Thus, transformants with the

desired allele swap can be easily identified by screening for the loss of the resistance or proto-trophy conferred by the cassette.

We analyzed the results of 40 independent allele exchanges we performed to determine the efficiency of CRISPR-Swap. After transformation with the CRISPR-Swap plasmid expressing either gCASS5a or gGFP and a PCR-generated repair template, a median of 87.5% of the transformants no longer expressed the cassette selectable marker. We screened over 100 of these transformants for integration of the desired allele by colony PCR, restriction digestion or sequencing and found that all had the correct allele exchange. We also sequenced the guide RNA recognition site in 13 transformants that remained G418 resistant and found that 2 had the hallmarks of repair by non-homologous end-joining, while the remaining 11 were repaired using the homologous sequence present in the GFP-His3MX cassette in these strains (see below) to repair the locus. We observed no difference in CRISPR-Swap efficiency among the two strain backgrounds (BY and RM), the genomic loci we targeted, or the gCASS5a and gGFP gRNAs. In summary, CRISPR-Swap readily creates allele replacements with high efficiency.

## Fine-mapping of hotspot regions using GFP-tagged proteins to measure *trans*-gene expression

We leveraged the ability of CRISPR-Swap to rapidly engineer allelic series at a given locus to dissect three *trans*-acting hotspot regions to the causal variant. Earlier computational fine-mapping had narrowed 26 hotspots in the BY/RM cross to three or fewer genes [12]. From among these 26 hotspots, we selected three for experimental dissection based on the availability of 1) a likely candidate gene given the enriched functions of the genes affected by the locus (see details below) and 2) an abundantly expressed gene that was strongly affected by the hotspot in *trans* and that showed a clear functional relationship to the other genes affected by this locus in *trans*. At each hotspot, we used the respective gene to monitor the effects of our genome edits on gene expression (Fig 1C). We C-terminally tagged the open reading frame of this gene with GFP, engineered the hotspot locus with CRISPR-Swap, and measured GFP fluorescence in each engineered strain during growth on a 96-well plate reader (Methods). This approach provided high-throughput measurements of gene expression for the statistically powerful dissection of the hotspot loci.

## A missense variant in *RGT2* influences expression of *HXT1* in trans

A hotspot locus on chromosome IV affects the expression of many genes at the mRNA and protein levels in crosses between BY and RM [12]. In the expanded BY/RM segregant panel, this locus had been mapped to a region containing three genes: *RGT2*, *ARF2*, and *RPL35-B* (Fig 2A). The mRNA levels of 1,400 genes are affected by this hotspot in *trans*, and these genes are enriched for roles in ATP synthesis and carbohydrate derivative metabolism. The strongest *trans* effect was on *HXT1* mRNA (LOD = 182), which encodes a low affinity glucose transporter whose expression is affected by Rgt2 [51], suggesting *RGT2* is likely the causal gene at this hotspot. In support of this, genetic variation within the *RGT2* coding region was previously shown to influence protein levels of the high affinity glucose transporter Hxt2p [29]. *RGT2* encodes a low affinity glucose sensor located in the plasma membrane (Fig 2B; reviewed in [52]). The BY and RM alleles of *RGT2* differ at five missense single nucleotide variant (SNVs). Three of these SNVs are located in the long cytoplasmic-facing C-terminal tail required for glucose signaling, and the remaining two are within the 12 predicted transmembrane helices (Fig 2B). *RGT2*, as well as *ARF2*, is also influenced by a local eQTL.

To determine if *RGT2* is the causal gene at this hotspot and identify the responsible variant, we C-terminally tagged Hxt1 with GFP and created a series of allele replacements at the *RGT2*

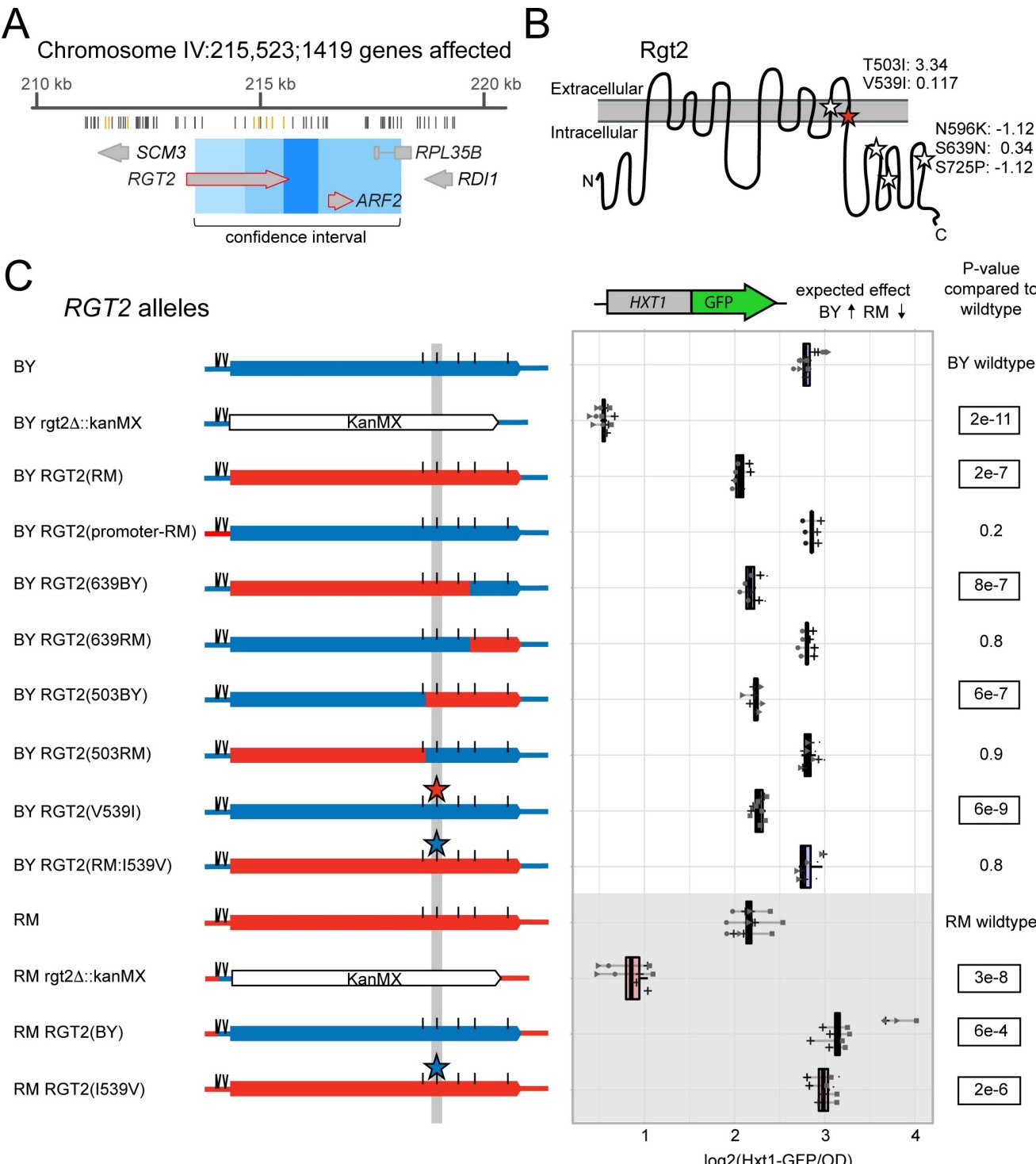

**Fig 2. Fine mapping the causal variant in *RGT2*.** A. Schematic of the eQTL hotspot on Chromosome IV. From top to bottom, the figure shows the hotspot location on the chromosome and the number of genes it affects, the positions of the BY and RM sequence variants (gray lines mark synonymous and intergenic variants and orange lines mark missense variants), the genes and their positions in the hotspot (large gray arrows; red outlines denote genes with a local eQTL) The 90% confidence interval (light blue), 95% confidence interval (medium blue) and the position of markers perfectly linked to the hotspot peak marker (dark blue). Modified from Supplementary File 5 in [12]. B. Schematic of the Rgt2 protein in the plasma membrane. BY / RM missense variants are indicated by stars, with the identified causal variant in red. The missense variant amino acid positions, their resulting amino acid change and their PROVEAN scores [53] are shown. The synonymous variants are not depicted. C. On the left are schematics of *RGT2* alleles with BY sequences in blue and RM sequences in red. Positions of SNVs are marked with a straight line and INDELs are marked with two diagonal lines. The

variants in the *RGT2* 3'UTR and synonymous variants are not depicted. On the right are the corresponding Hxt1-GFP fluorescence levels for each allele. The direction of the expected effect on Hxt1-GFP from [12] is shown. P-values are for tests comparing each allele to its respective wildtype. Significant p-values are outlined. Blue boxplots indicate alleles in the BY background and red boxplots and gray background shading indicate alleles in the RM background. Lines group measurements of the same clone. Different symbols (circles, squares, etc.) denote different plate reader runs. Small dots next to a clone indicate a 5'UTR indel with an allele that does not match the one indicated in the allele schematics (see Methods).

locus using CRISPR-Swap. In both BY and RM backgrounds, deletion of *RGT2* drastically reduced Hxt1-GFP levels, confirming that *RGT2* is required for proper Hxt1-GFP expression. Replacing the BY *RGT2* promoter region with the RM allele had no measurable effect on Hxt1-GFP. Reciprocal allele replacement of the entire *RGT2* coding region showed that the *RGT2* RM allele resulted in lower Hxt1-GFP expression compared to the BY allele (Fig 2C), which is the direction of effect expected from eQTL data [12].

By engineering a series of chimeric *RGT2* alleles spanning the coding region we systematically narrowed the causal region to two missense variants, neither of which were predicted to be deleterious (Fig 2C; S1 Table). Of these, a G-to-A SNV at 241,965 bp, resulting in a valine to isoleucine change at amino acid position 539 (V539I), recapitulated the effect of swapping the entire coding region. In a strain that carried all *RGT2* RM coding variants *except* V539I, Hxt1-GFP expression was indistinguishable from the BY wildtype, suggesting that V539I is the single causal variant in this gene. The effect of V539I on Hxt1-GFP expression was present in both BY and RM (Fig 2C), with no evidence that the strain background influences the effect of this variant (interaction p-value = 0.1). The variant caused a minor effect on growth rate in the RM background, but no effect in the BY background (Table 1).

## The *RGT2(V539I)* variant affects Hxt1-GFP expression in a glucose-dependent manner

Increasing glucose concentrations results in an increase in the expression of Hxt1 [51]. Given that the V539I variant lies within one of the predicted transmembrane helices that likely form the pocket necessary for glucose sensing by Rgt2 [54], we hypothesized that the effect of the V539I variant may change depending on the concentration of glucose in the culture medium. To test this idea, we measured Hxt1-GFP expression in BY and RM strains with their native *RGT2* alleles as well as with engineered V539I alleles in a range of glucose concentrations (Fig 3).

In both BY and RM, higher glucose concentrations increased expression of Hxt1-GFP regardless of which V539I allele was present. However, the difference in Hxt1-GFP expression between the V539I alleles showed a clear dependence on glucose levels (ANOVA test for interaction between allele effect and glucose concentration: p = 7e-13). The BY allele drove nearly 2-fold higher Hxt1-GFP expression than the RM allele at 1% glucose, while the difference between alleles became almost indistinguishable at 12% glucose. The *RGT2* hotspot had been mapped in YNB medium with 2% glucose [12], which was among the concentrations at which the V539I variant showed a large effect (Fig 3). Thus, the effect size of the V539I variant

**Table 1. Effects of causal hotspot variants on growth rates.**

| Variant | Growth rate in BY (RM allele relative to BY wildtype allele) | p-value in BY | Growth rate in RM (BY allele relative to RM wildtype allele) | p-value in RM |
|---|---|---|---|---|
| *RGT2 (V539I)* | 98% | 0.5 | 93% | 0.02 |
| *OAF1(S63L)* | 96% | 0.09 | 100% | >0.99 |
| *OLE1(FAR)* | 92% | 0.3 | 92% | 0.005 |

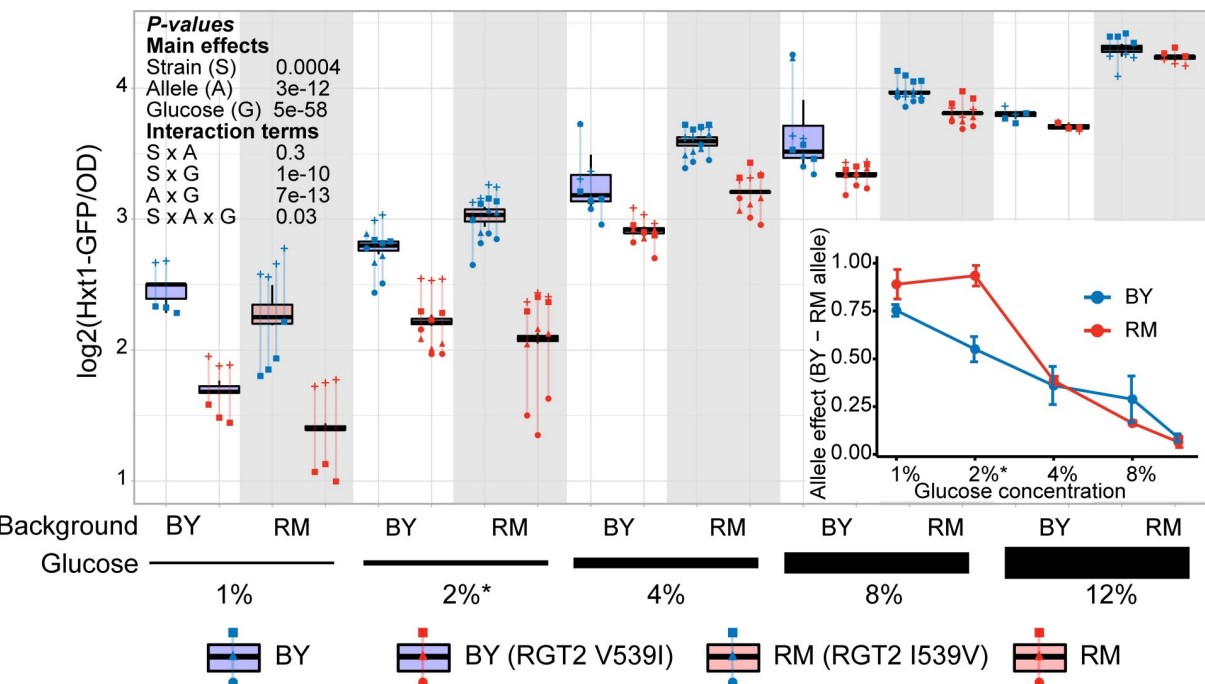

**Fig 3. Effect of the *RGT2(I539V)* variant on Hxt1-GFP expression in different glucose concentrations.** Hxt1-GFP fluorescence levels in the four different genotypes in increasing glucose concentrations are shown. The asterisk indicates the glucose concentration used when identifying this eQTL. Blue boxplots indicate alleles in the BY background and red boxplots and gray background shading indicate alleles in the RM background. Lines group measurements of the same clone. Different symbols (circles, squares, etc.) denote different plate reader runs. The inset graph shows fitted effect sizes as a function of glucose concentration. Error bars show standard errors of the mean. P-values are shown for ANOVA models examining the various main and interaction terms.

underlying the *RGT2* eQTL hotspot strongly depends on the environment, and the initial discovery of this hotspot may have been aided by common yeast media conditions that result in a large effect at this locus.

## A missense variant in *OAF1* alters expression of *FAA4* in trans

Several of the eQTL hotspots identified in the BY/RM cross affect the expression of genes involved in fatty acid metabolism. They include a hotspot on chromosome I, which affects the mRNA and protein levels of 450 genes in *trans*. The locus was previously mapped to a region containing the genes *FLC2* and *OAF1* [12] (Fig 4A). Neither *FLC2* nor *OAF1* are influenced by a local eQTL, suggesting the causal variant is likely coding. *OAF1* encodes a transcription factor that activates expression of genes involved in peroxisome related functions including the β-oxidation of fatty acids [55–57], making it a promising candidate causal gene [58]. The BY and RM alleles of *OAF1* differ at three missense SNVs (Fig 4B).

We engineered a series of *OAF1* alleles and measured Faa4-*GFP* expression in *trans*. *FAA4* mRNA abundance was shown to be strongly affected by this locus (LOD score = 102, [12]), and *FAA4* encodes a long chain fatty acyl-CoA synthetase, a protein with clear phenotypic connection to fatty acid metabolism. We found that a single T-to-C missense variant in *OAF1* at 48,751 bp resulted in a decrease in Faa4-GFP expression in agreement with the direction of effect observed in the eQTL data (Fig 4C, S2 Fig, S1 Table). The variant encodes a leucine at position 63 in BY and a serine in RM (L63S). In the Oaf1 protein, the L63S variant is located next to the Zn(2)Cys(6) DNA binding domain (Fig 4B) where it may alter Oaf1 binding to its regulatory targets. The effect of L63S on gene expression was consistent in both strain

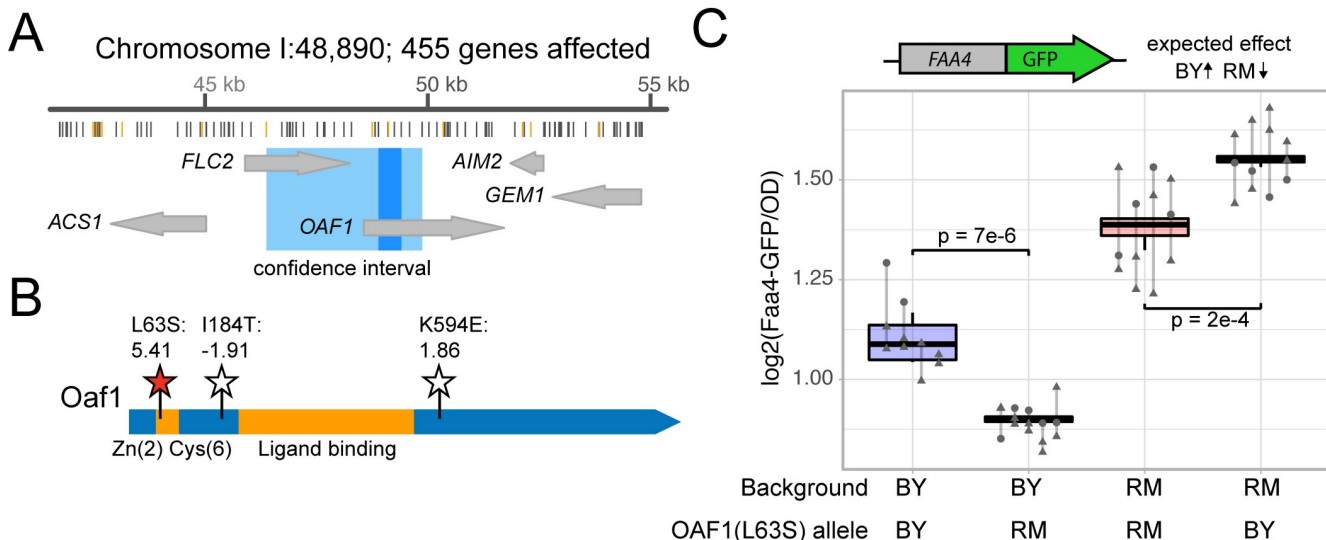

**Fig 4. The *OAF1* causal variant.** A. Schematic of the eQTL hotspot on Chromosome IV. From top to bottom, the figure shows the hotspot location on the chromosome and the number of genes it affects, the positions of the BY and RM sequence variants (gray lines mark synonymous and intergenic variants and orange lines mark missense variants), the genes and their positions in the hotspot (large gray arrows), the 95% confidence intervals (medium blue) and the position of markers perfectly linked to the hotspot peak marker (dark blue). For this locus, the 95% and 90% confidence intervals were identical. Modified from Supplementary File 5 in [12]. B. Schematic of the Oaf1 protein domains and variants. The conserved zinc-finger domain and the ligand-binding domain [59] are in orange. BY / RM missense variants are indicated by stars, with the identified causal variant in red. The missense variant amino acid positions, their resulting amino acid change and their PROVEAN scores [53] are shown. The synonymous variants are not depicted. C. The figure shows Faa4-GFP fluorescence levels for strains with the indicated *OAF1(L63S)* alleles. The direction of the expected effect on Faa4-GFP from [12] is shown. P-values are for tests comparing the allele to its respective wildtype. Blue boxplots indicate alleles in the BY background and red boxplots indicate alleles in the RM background. Lines group measurements of the same clone. Different symbols (circles, squares, etc.) denote different plate reader runs.

backgrounds (interaction p-value = 0.4; Fig 4C), and L63S had no detectable effects on growth rate (Table 1).

## A noncoding promoter variant influences *OLE1* expression in *cis* and *FAA4* in *trans*

A second fatty-acid related hotspot resides on chromosome VII and affects the mRNA levels of 977 genes, which are enriched for functions in lipid biosynthetic processes as well as the response to endoplasmic reticulum stress. The confidence interval of this locus spans ~1 kb centered in the noncoding region between the *SDS23* and *OLE1* genes (Fig 5A). *OLE1* encodes the ER-bound Δ9-fatty acid desaturase, the only yeast enzyme capable of desaturating fatty acids [60,61]. *SDS23* is likely involved in regulation of the metaphase to anaphase transition of the cell cycle [62]. Given the enrichment of target genes of this hotspot involved in lipid metabolism, we reasoned that *OLE1* is the more likely causal gene. The BY and RM alleles of *OLE1* differ at one missense SNV and four non-coding variants (2 indels and 2 SNVs) in the *SDS23* and *OLE1* intergenic region.

We engineered the essential *OLE1* locus using double-cut CRISPR-Swap, by flanking the region with the HphMX and KanMX cassettes and replacing both cassettes along with the intervening region with a series of alleles (Fig 1B). We again measured Faa4-GFP expression in the engineered strains, given *FAA4* mRNA levels are strongly affected by this locus (LOD = 78). We identified a noncoding A-to-G SNV at 398,081 bp in the intergenic region between *SDS23* and *OLE1* that affected Faa4-GFP expression in the expected direction based on the eQTL data (S3 Fig, S1 Table). This effect was consistent in both strain backgrounds (interaction p-value = 0.8). While the BY allele of this variant decreased growth rate in the RM background, the variant had no effect in the BY background (Table 1).

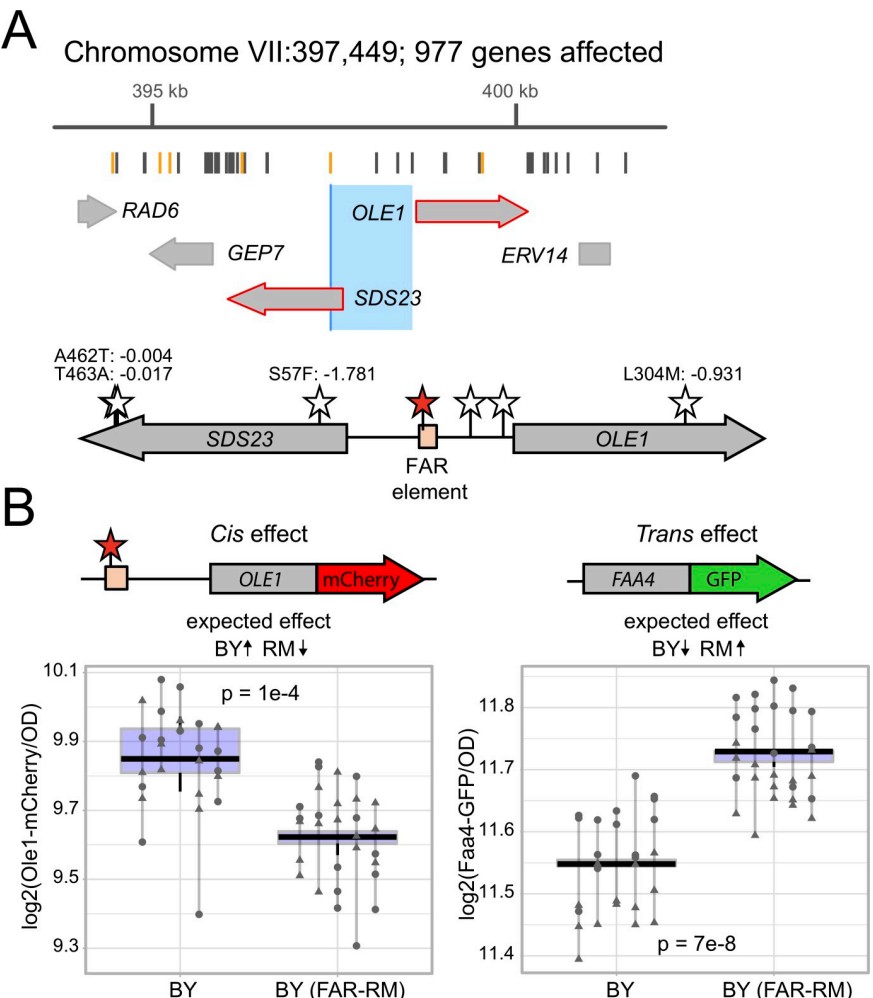

**Fig 5. The *OLE1* causal variant.** A. Schematic of the eQTL hotspot on Chromosome VII. From top to bottom, the figure shows the hotspot location on the chromosome and the number of genes it affects, the positions of the BY and RM sequence variants (gray lines mark synonymous and intergenic variants and orange lines mark missense variants), the genes and their positions in the hotspot (large gray arrows; red outlines denote genes with a local eQTL) The 95% confidence interval (medium blue) and the position of markers perfectly linked to the hotspot peak marker (dark blue). For this locus, the 95% and 90% confidence intervals were identical. Modified from Supplementary File 5 in [12]. B. Schematic of the *SDS23* / *OLE1* region. BY / RM variants in the intergenic region and missense variants in the genes are indicated by stars, with the identified causal variant in the FAR element (peach box) in red. The missense variant amino acid positions, their resulting amino acid change and their PROVEAN scores [53] are shown. The synonymous variants are not depicted. C. Fluorescence levels of Ole1-mCherry (left panel) and Faa4-GFP (right panel). The BY strains harbor both Ole1-mCherry and Faa4-GFP and the indicated *OLE1* alleles. The direction of the expected effects on Ole1-mCherry and Faa4-GFP from [12] are shown. Lines group measurements of the same clone. Different symbols (circles, squares, etc.) denote different plate reader runs.

The causal variant is located in the fatty-acid regulated (FAR) promoter element, a region known to be important for transcriptional activation and fatty acid regulation of *OLE1* expression [63] (Fig 5B). Both *OLE1* and *SDS23* are strongly affected by local eQTLs with higher mRNA expression linked to the BY allele, suggesting that *OLE1* and/or *SDS23* expression may be affected by the FAR variant. To test the effect of the FAR variant on *OLE1* expression, we created BY strains with *OLE1* tagged with mCherry in addition to *FAA4* tagged with *GFP*. In these strains, the RM FAR allele significantly decreased Ole1-mCherry levels in *cis* and increased Faa4-GFP levels in *trans* (Fig 5C), which are the directions of effect expected based on the eQTL results [12].

To further examine the *cis*-regulatory activity of the FAR variant, we created reporter plasmids with multiple alleles of the *SDS23/OLE1* intergenic region driving expression of yeVenus. When the intergenic region drove expression of yeVenus in the direction of *SDS23*, the promoter alleles all drove similar yeVenus expression (S4 Fig). In sharp contrast, when the intergenic region drove expression of yeVenus in the direction of *OLE1*, the FAR-BY allele drove significantly higher yeVenus expression than the RM allele, fully recapitulating the difference between the all-BY and all-RM promoter alleles (S4 Fig). The effect did not depend on whether the yeVenus plasmids were expressed in the BY or RM background (interaction p > 0.29). Together, these results suggest that the FAR variant influences *OLE1* in *cis*, as well as *FAA4* in *trans*.

## Small changes in *OLE1* expression from wildtype levels are sufficient to affect *FAA4* gene expression in *trans*

To explore the effect of *OLE1* and *SDS23* abundance on gene expression in *trans*, we added an additional copy of *OLE1*, *SDS23* or the intergenic sequence on a low-copy plasmid in a BY *FAA4-GFP* strain. The presence of an extra copy of *OLE1* resulted in significant reduction of Faa4-GFP levels, consistent with eQTL data (S5 Fig). By contrast, an extra copy of *SDS23* resulted in an increase in Faa4-GFP levels. An extra copy of the intergenic sequence alone did not change Faa4-GFP levels (S5 Fig). These results further support that it is a change in *OLE1* and not *SDS23* expression that is responsible for the *trans*-effect of the hotspot on *FAA4* expression.

While changes in Ole1 levels appear to be the primary cause for the *trans* effect on Faa4-GFP, the noncoding FAR variant alters *OLE1* expression by only about 15% (Fig 5B). To understand the relationship between *OLE1* and *FAA4* over a range of expression levels, we inserted a synthetic, inducible $Z_3EV$ promoter upstream of *OLE1*. This promoter can be activated precisely and quantitatively by addition of estradiol to the culture medium [64]. We measured both Ole1-mCherry and Faa4-GFP as a function of estradiol concentration and observed a clear anticorrelation between Ole1-mCherry and Faa4-GFP levels (Fig 6). At an estradiol dose of 4–5 nM, the $Z_3EV$ strain expressed Ole1-mCherry and Faa4-GFP at levels comparable to BY strains expressing *OLE1-mCherry* from its native promoter with either the wildtype or FAR-RM allele. Higher doses of estradiol continued to increase Ole1-mCherry levels, while Faa4-GFP dropped rapidly and reached a plateau at levels well below the native BY strains. As expected for the essential *OLE1* gene, low levels of induction resulted in poor growth (S6 Fig). Taken together, our results indicate that the BY strain expresses *OLE1* at a level at which even slight alterations in *OLE1* expression, like that caused by the FAR variant, are sufficient to change *FAA4* expression in *trans*.

## A non-additive interaction between the *OAF1* and *OLE1* variants

The Oaf1 transcription factor binds throughout the *OLE1* promoter including in the close vicinity of the FAR variant [65]. This raises the possibility that the effects of *OAF1(L63S)* and the FAR variant may interact with each other genetically. While neither variant had shown significant interactions with the genetic background as a whole (see above), a specific interaction between these two variants could have been obscured by the thousands of other sequence differences between the BY and RM backgrounds. Indeed, the previous eQTL data contained a non-additive interaction affecting *FAA4* mRNA levels between two *trans* eQTLs that contained *OAF1* and *OLE1*, respectively (Fig 7A). To test if this *trans*-by-*trans* interaction could be explained by the specific causal variants we identified here, we constructed a BY strain that carried the RM alleles at both *OAF1* and the FAR variant and compared its Faa4-GFP

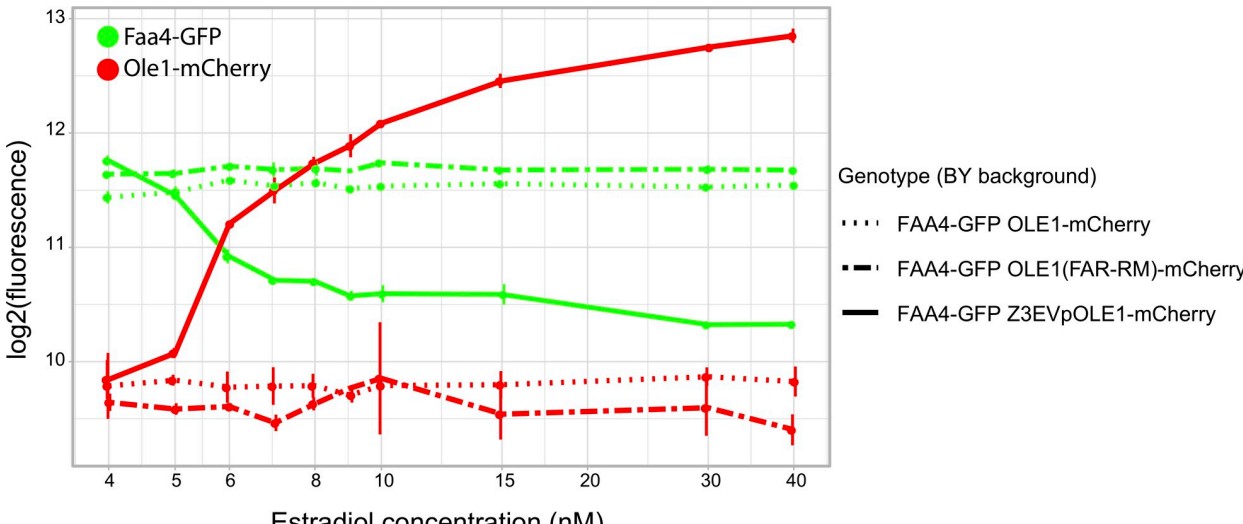

**Fig 6. Expression modulation of Ole1.** Fluorescence levels of Ole1-mCherry (solid red lines) under control of the estradiol-inducible Z₃EV promoter and Faa4-GFP (solid green lines) under its native promoter at various estradiol doses. For comparison, fluorescence levels of strains with the native *OLE1* promoter with the FAR-BY or FAR-RM allele are shown as dotted and dashed lines, respectively. All strains harbor both the mCherry tag on Ole1 and the GFP tag on Faa4. Measurements are shown as mean (dots) and standard deviations (vertical lines).

expression to strains that carried one or the other of these edits. The combined alleles resulted in Faa4-GFP expression that differed from an additive allele combination (interaction p = 0.002) in a manner that mirrored the detected eQTL interaction (Fig 7B). Specifically, in the presence of the more active RM *OAF1* allele, the FAR variant showed a greater effect than in the presence of the less active BY *OAF1* allele. As with the majority of epistatic interactions

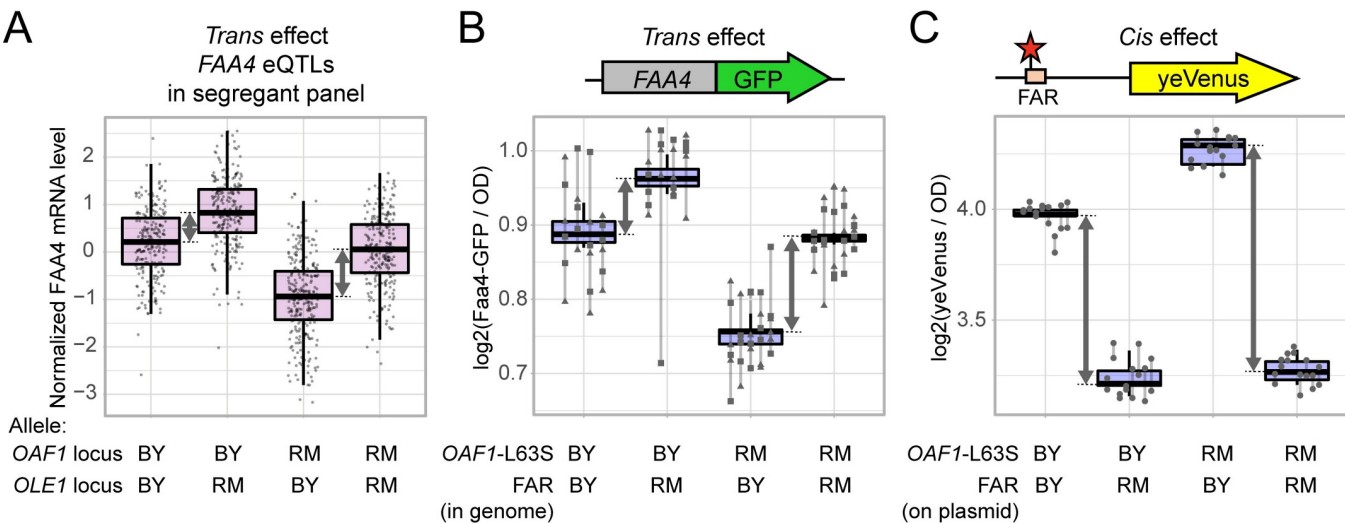

**Fig 7. Nonadditive genetic interaction between the *OAF1* and *OLE1* causal variants.** A. Data from [12] show levels of *FAA4* mRNA in 1,012 BY/RM segregants, divided into four groups depending on their genotypes at *OAF1* and *OLE1*. Note the slightly larger difference between *OLE1* alleles in segregants that carry the RM allele compared to the BY allele at *OAF1*. B. Faa4-GFP fluorescence levels in strains in the BY background engineered to carry the BY or RM allele at *OAF1(L63S)* and *OLE1(FAR)*. C. yeVenus fluorescence levels for strains in the BY background engineered to carry the BY or RM allele at *OAF1(L63S)* and the *OLE1(FAR)* promoter driving yeVenus expression on a plasmid. The difference between *OLE1* alleles in segregants (in A) or strains (in B and C) that carry the RM allele compared to the BY allele at *OAF1* are highlighted by gray arrows.

identified in this cross [66], the deviation from additivity caused by this interaction is detectable but subtle.

To test if this *trans*-by-*trans* interaction affecting *FAA4* could be mediated by a *cis*-by-*trans* interaction between the same two variants, we introduced our yeVenus plasmids with the two FAR alleles into BY strains with either the BY or RM *OAF1(L63S)* alleles. We again found a significant interaction between the FAR and *OAF1* variants (Fig 7C). In the presence of the *OAF1(L63S)*-RM allele, the FAR variant resulted in a larger difference in yeVenus expression than in the presence of the *OAF1(L63S)*-BY allele (p = 1.5 e-7). The *OAF1* RM allele also resulted in overall higher expression of the yeVenus construct (p = 1.6 e-6). Taken together, these results show that the *OAF1(L63S)* variant and the *OLE1* FAR variant interact genetically such that their effects on gene expression in *cis* as well as *trans* are not simply additive.

### The *OAF1* and *OLE1* variants alter lipid profiles

To test if the two causal variants in the fatty acid metabolism genes *OAF1* and *OLE1* alter cellular phenotypes other than gene expression, we measured overall lipid composition as well as non-esterified ("free") fatty acids (NEFAs) in the RM and BY strains, as well as in BY strains with *OAF1(L63S)*-RM, *OLE1(FAR)*-RM, or both of these alleles (Fig 8 & S7 Fig). The BY and RM strains differed in multiple metabolites (S7 Fig, S2 Table, S3 Table), and the presence of either of the two variant alleles in the BY background also resulted in significant differences in lipid metabolites. The BY *OAF1* allele decreased the fraction of longer-chain (C18) lipids (Fig 8A) and also caused a decrease in the amount of C18 NEFAs (S7 Fig). This change resembles the effect of an *OAF1* deletion on lipid metabolism [65], consistent with the BY *OAF1* allele having reduced function (S2 Fig). The *OLE1(FAR)*-RM allele resulted in a significant increase in the amount of saturated NEFAs (Fig 8B), as expected if reduced expression of the Ole1 desaturase caused by the FAR-RM allele results in reduced Ole1 activity. Conversely, there was not a significant reduction in the amount of desaturated NEFAs, suggesting that their cellular

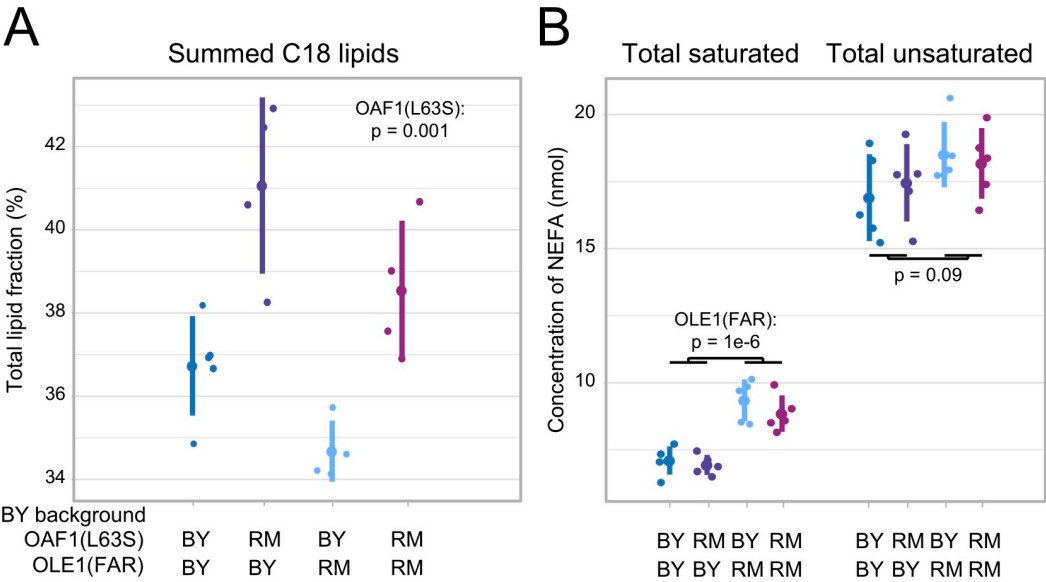

**Fig 8. Lipid and fatty acid measurements.** A. The fraction of C18 lipids in BY strains with wildtype or RM alleles at *OAF1 (L63S)* or *OLE1(FAR)* or a combination of both. B. The concentration of saturated and unsaturated non-esterified fatty acids (NEFA) in the same strains. For each strain, the figure shows values for each replicate (smaller points) along with the mean (larger points) and standard deviation (vertical lines). See S7 Fig for all measurements.

levels are not solely dependent on Ole1 levels. We did not detect changes in overall lipid composition due to the *OLE1(FAR)*-RM allele (S7 Fig). Taken together, our results show that the variants affecting *OLE1* expression and Oaf1 activity translate to changes in cellular lipids.

## The fine-mapped variants are the cause of the hotspot effects on mRNA levels

We have shown that each of the variants in *RGT2*, *OAF1*, and the *OLE1* FAR element affect the protein expression of a representative gene in *trans*. If these variants underlie the *trans*-acting hotspots, we expect them to alter the transcript levels of many genes, and that these expression changes will correlate with the known effects of the hotspots we sought to fine-map. To test this, we quantified transcript levels in BY strains edited at each variant (S4 Table–S6 Table).

Each of the three variants altered the transcript levels of dozens of genes including the representative gene we used for fine-mapping (*HXT1* or *FAA4*) with the expected direction of effect (Fig 9, Table 2). In addition, the FAR variant caused a nominally significant (p = 0.03) effect on *OLE1* but not on *SDS23* (p = 0.7), in agreement with our yeVenus reporter assay (S4 Fig). Crucially, the magnitude of expression change caused by the three variants was significantly and positively correlated with the respective hotspot effects (Fig 9) when considering all expressed genes. Like the vast majority of *trans* eQTLs [12], the three hotspots dissected here have small effects on most genes, typically explaining only a few percent of variance in mRNA levels. Our RNA-Seq experiment was not designed to detect such small effects at statistical significance, which would require dozens to hundreds of replicates. When we used a lenient significance cutoff (uncorrected p < 0.05) to restrict our analysis to genes with some evidence for differential expression, the correlations with hotspot effects increased at each hotspot (Table 2). These strong correlations were reflected in high directional concordance. For example, at a more stringent threshold (false discovery rate = 10%), every differentially expressed gene with a hotspot effect had concordant direction of effect with the given hotspot for *RGT2* and *OLE1*, and there was just a single discordant gene (out of 30) for *OAF1* (Fig 9). This strong

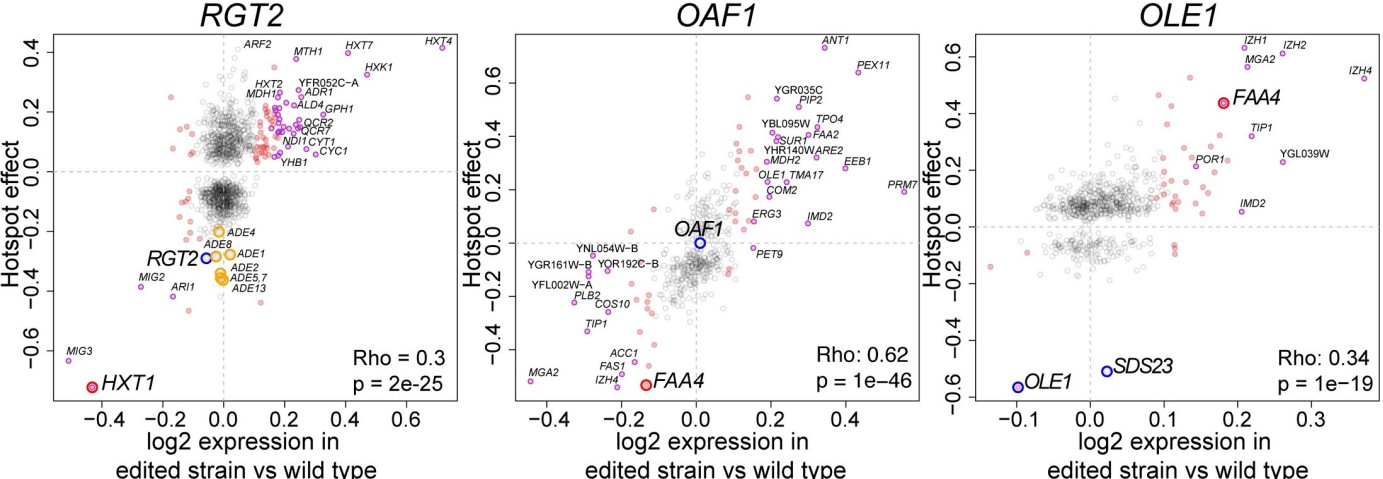

**Fig 9. Comparison of the differential gene expression caused by each causal variant and the eQTL hotspot effects.** The differential expression of an edited vs wildtype strain is on the x-axis and the previously determined hotspot effects are on the y-axis. Hotspot effects are coefficients from a regularized linear regression of expression values on the genotype at the respective hotspot marker [12]. Genes with non-zero hotspot effects are shown. Red points are genes with differential expression p-value < 0.05. Purple points are genes with differential expression at FDR < 0.1. The larger red circles mark the genes used for fine-mapping. The blue circles are genes we investigated in the confidence interval of each hotspot. The correlation between differential expression and hotspot effects is given in each panel. For *RGT2*, orange circles show adenine-metabolism related genes that are described in the text.

**Table 2. RNA Seq results.**

| Variant | Number of DE genes (10% FDR) | Number of DE genes (p < 0.05) | Correlation[1] between DE[2] and hotspot effect[3]: genes with DE p < 0.05 | Percent of DE genes (FDR <10%) with a hotspot effect; odds ratio[4] and p-value[4] |
|---|---|---|---|---|
| *RGT2 (V539I)* | 45 | 183 | Rho = 0.64, p ≈ 0 | 86%; OR = 24, p<2.2e-16 |
| *OAF1 (L63S)* | 45 | 147 | Rho = 0.80, p = 6e-15 | 71%; OR = 31, p<2.2e-16 |
| *OLE1 (FAR)* | 18 | 95 | Rho = 0.48, p = 0.0015 | 50%; OR = 7, p = 0.0001 |

[1]These correlations exclude genes without a hotspot effect detected in [12].

[2]log2 fold change

[3]Hotspot effects are coefficients from a regularized linear regression of expression values on the genotype at the respective hotspot marker [12].

[4]Fisher's exact test for association between differential expression and a non-zero hotspot effect

agreement between expression changes caused by the three variants and known hotspot effects shows that these variants are causal variants at their respective hotspots.

If a hotspot is caused by multiple causal variants in the same or neighboring genes, our single variant edit might not account for all the effects of the hotspot. Therefore, we examined genes with transcript levels strongly affected by the hotspot that are unaffected by our variant edits. Genes strongly affected by the hotspot at *RGT2* but not by the *RGT2(V539I)* variant showed a significant enrichment for "*de novo* IMP biosynthetic process" (corrected p = 9e-9) and related terms in purine metabolism. This enrichment is driven by seven ADE genes with large hotspot effects that were all non-significant in our experiment (Fig 9). We suspect that a second variant in this region is responsible for these effects.

## Population distribution and conservation of causal variants

To explore the evolutionary history of the three causal hotspot variants, we examined their distribution across a worldwide panel of 1,011 *S. cerevisiae* isolates with genomic sequence [67]. The BY allele at *OAF1(L63S)* is rare among yeast isolates (Fig 10). It is carried only by BY and a few close relatives while the RM allele is present in all other isolates as well as related species. Reflecting this pattern, the Protein Variation Effect Analyzer (PROVEAN) tool [53] assigned a "deleterious" score of -5.4 to the BY allele at this variant. In our experiments, the BY *OAF1* allele increased Faa4-GFP expression, which was the same direction caused by the *OAF1* knockout (S2 Fig). Thus, the *OAF1* hotspot is caused by a rare, derived missense variant almost exclusive to the BY laboratory strain that probably reduces function of the Oaf1 transcription factor.

The *RGT2* and *OLE1* variants show very different patterns compared to *OAF1(L63S)*. At *RGT2(V539I)*, both the valine in BY and the isoleucine in RM are also encoded by related yeast species (Fig 10). Evidently, the V539I variant can be tolerated without severe fitness consequences, as reflected in a "neutral" PROVEAN score of 0.12. Within *S. cerevisiae*, the highly divergent and likely ancestral group of Chinese isolates [67–69] carry the valine found in BY, suggesting that the isoleucine in RM is derived. This derived allele has ~25% frequency in the *S. cerevisiae* population, where it is predominantly found in isolates from the European wine clade, as well as in a second group of isolates with mixed origin (Fig 10). Both the derived RM variant and an *RGT2* deletion resulted in reduced induction of *HXT1* expression (Fig 2C).

At the *OLE1(FAR)* variant, the alanine found in BY is present in the ancestral Chinese isolates, suggesting that the guanine in RM is derived (Fig 10). Indeed, the nucleotide sequence of

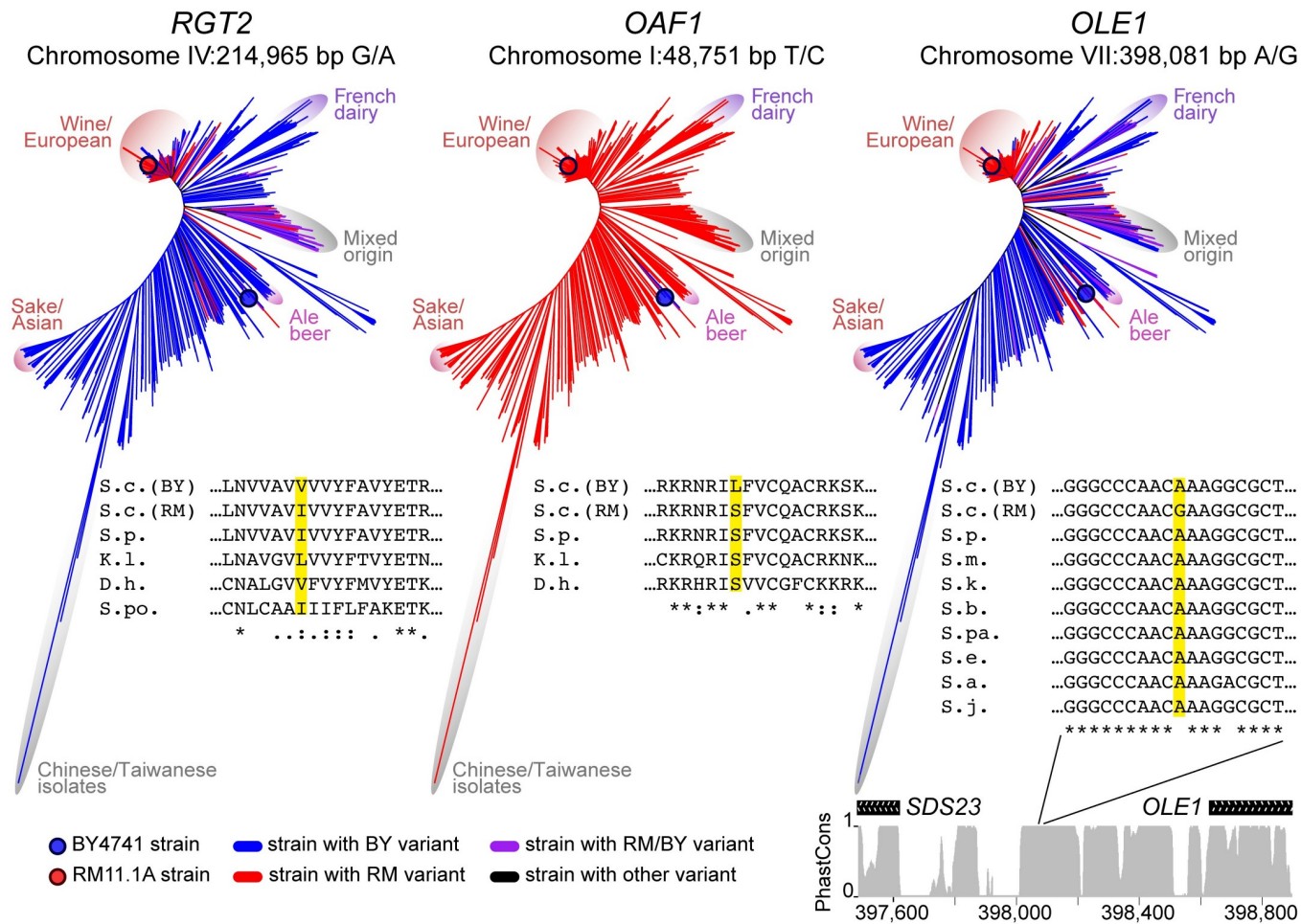

**Fig 10. Population genetic features of the causal variants.** Neighbor joining tree of the 1,011 *S. cerevisiae* strains sequenced in [67]. For each causal variant, the presence of the BY (blue), RM (red), BY and RM (purple), or other (black) allele is shown across the tree. For the *OAF1* and *RGT2* missense variants, amino-acid alignments with the indicated species are shown. For the intergenic FAR variant, we show nucleotide alignments as well as PhastCons conservation scores for the depicted *SDS23 / OLE1* region from the UCSC Genome Browser. In alignments, species names are abbreviated: S.c., *Saccharomyces cerevisiae*; S. p., *S. paradoxus*; S.m., *S. mikatae*; S.k., *S. krudriavzevii*; S.b., *S. bayanus*; S.pa., *S. pastorianus*; S.e., *S. eubaynus*; S.a., *S. arboricola*; S.j., *S. jurei*; K.c., *Kluyveromyces lactis*; D.h., *Debaryomyces hansenii*; S.po., *Schizosaccharomyces pombe*.

the noncoding region surrounding the FAR variant is conserved among *Saccharomyces sensu stricto* species, and all other species carry the alanine found in BY at this position. The RM allele has high frequency (46%) among yeast isolates, predominantly due to near fixation among the many isolates in the European wine clade. The allele is also present in isolates from dairy, ale beer, and other origins. The RM allele increased *FAA4* expression in *trans*. This is the same direction of effect we observed when inserting the kanMX cassette immediately downstream of *OLE1*. Such engineered alleles are commonly called "Decreased Abundance by mRNA Perturbation" (DAmP) alleles and are expected to decrease gene function by lowering mRNA transcript levels ([70]; S3 Fig).

Thus, the three causal variants identified here have diverse population genetic characteristics. They include a rare, lab-specific allele and two common alleles found in a quarter or more of the sequenced yeast isolates. At all three variants, we observed that the likely derived allele altered gene expression in the same direction as alleles that eliminate or reduce gene function.

## Discussion

We fine-mapped natural DNA variants that each result in expression changes at many genes in *trans* using CRISPR-Swap, a strategy that facilitates rapid engineering of allelic series at a given locus. CRISPR-Swap is similar to other recently developed two-step engineering approaches (e.g. [71–74]) and has multiple advantages: 1) The great majority of clones that are transformed with the CRISPR-Swap plasmid and repair template incorporate the desired allele and clones without the desired allele are easily identified by screening for maintained expression of the cassette selectable marker. 2) gRNAs do not need to be designed and tested for each region due to the use of a common gRNA. 3) Regions without a nearby PAM site can be engineered because the gRNAs target integrated cassettes rather than the genomic region directly. 4) Larger regions, including those that contain essential genes can be engineered by flanking the region with two cassettes and using a single gRNA to cut and swap both cassettes and the intervening region. 5) Our gRNAs can be directly used to engineer existing strains that already contain cassettes e.g., strains in the *S. cerevisiae* deletion and GFP collections [75,76].

While we successfully used GFP-tagged protein abundance as phenotypic readouts amenable to high-throughput measurement, the hotspots we dissected had been identified via their effects on mRNA levels. The effects of the locus on the mRNA and protein of the gene used for phenotyping cannot always be assumed to be consistent [22,37]. Further, fine-mapping using the expression of a single focal gene can only detect variants that influence this focal gene. Additional variants in the same region that specifically affect other genes would be missed. Indeed, while our RNA-Seq results were consistent with single causal variants at *OAF1* and *OLE1*, they suggested the presence of a second causal variant close to *RGT2*, which acts on genes involved in purine metabolism. Evidently, even such narrowly mapped hotspot loci as those we dissected here can be due to multiple causal variants with distinct effects but in close proximity to each other, as has been observed for QTLs for other traits in yeast [77,78].

A key result from this work is that the three causal variants we identified differ from each other in several aspects. First, the variants include two coding missense variants (in *RGT2* and *OAF1*), along with the *cis*-acting noncoding FAR variant at *OLE1*. In yeast, 10 additional natural variants have been experimentally demonstrated to affect gene expression in *trans*, and there are 5 additional hotspots for which the gene but not the causal variant is known (S7 Table, [18,20,23,30,31,34,38–41], see also [79]) Coding variation underlies at least 13 of these 18 cases, including missense, frameshift and transposon insertion variants. In species other than yeast, information about causal *trans* eQTL variants remains extremely limited [80–83]. In human genetics, searches for *trans* eQTLs often assume a model in which noncoding variants alter the expression of a regulator gene in *cis*, which in turn alters the expression of other genes in *trans* [84–88]. The FAR variant at *OLE1* discovered here is an example of such a mechanism. While the predominance of causal coding variants in yeast may in part reflect the density of coding regions in the yeast genome, it seems likely that *trans* eQTLs caused by coding variants may also exist in other species.

Second, the genes affected by the three variants encode different types of proteins: a glucose sensor (Rgt2), a transcription factor (Oaf1), and the essential enzyme Ole1. While genes encoding transcription factors are enriched in hotspot regions [12], hotspots clearly also arise from other gene classes (S7 Table). Among these, enzymes are a particularly interesting group. Because most enzymes do not directly regulate gene expression, metabolic changes caused by differential enzyme activity or expression must trigger *trans* changes in gene expression indirectly [89,90]. The FAR variant at *OLE1* illustrates the indirect mechanisms that could underlie such indirect *trans* effects. Its RM allele reduced Ole1 expression and increased saturated NEFAs. Higher lipid saturation decreases membrane fluidity [91–93] which is sensed by

membrane-bound dimers of Mga2 or Spt23 [91,94–96]. In our data, the FAR RM allele increased *MGA2* expression in *trans*, and some of the genes most strongly affected by the FAR variant are known Mga2 targets (including *FAA4* as well as *IZH2* and *IZH4*; Fig 9; [97]). Apparently, this noncoding variant perturbs mechanisms involved in membrane homeostasis, which may ultimately alter gene expression via the transcriptional regulator *MGA2*.

The *trans* effects of the FAR variant were caused by a decrease in Ole1 levels of only about 15% (Figs 5B & 6). Such sensitivity to small expression changes may be unusual among genes. While the BY and RM strains carry thousands of local eQTLs affecting at least half of the genes in the genome, most of these local eQTLs do not result in detectable expression changes at other genes in *trans* [12]. Little is known about whether, when, and how small expression changes at one gene influence other genes in *trans*, and how some of these changes go on to influence the organism. Pioneering studies have shown that even relatively small reductions in the expression of the enzyme genes *TDH3* [98] and *LCB2* [99] can reduce fitness, and that the relationship between gene expression and fitness is specific to each gene [100], the environment [100], and the strain background [99]. Future work will explore the causal relationships among fitness and gene expression changes in *trans*.

Finally, the three causal variants we discovered also differed in their population genetic characteristics. The *OAF1(S63L)* variant is rare, while the *RGT2* and *OLE1* variants are found in many isolates. These two common variants differ in the degree of evolutionary conservation of their site, with poor conservation at *RGT2(V539I)* and high conservation at the noncoding FAR variant. At least in the case of *RGT2*, simple evolutionary conservation alone would not have been sufficient to predict the causal variant.

At all three variants, the derived alleles affected *trans* gene expression in the same direction as loss of function or reduced function alleles. While this suggests that the derived alleles are detrimental, the high frequency in the population of the *RGT2* and *OLE1* alleles argues against strong negative fitness consequences. While these two variants could potentially be beneficial in some backgrounds or conditions, they could also be sufficiently mildly deleterious to have drifted to high frequency, as has been proposed to be the case for many *trans*-acting polymorphisms [101,102]. Indeed, none of the three variants resulted in consistent growth differences in both backgrounds in our culture medium (Table 1), suggesting that any fitness effects they may have are minor or occur in other environments. For example, at *RGT2*, the effect of the V539I variant on *HXT1* expression was reduced dramatically simply by altering the amount of glucose in the medium, suggesting that the long-term evolutionary consequences of this variant could be highly dependent on the environment.

With the arrival of population-scale genome sequencing [103], the functional interpretation of individual DNA variants has become a major goal of genetics, and numerous experimental and computational approaches aiming to predict phenotypic consequences of variants have been developed [53,104–111]. The diverse nature of variants that cause *trans*-eQTLs revealed here will make prediction of *trans* effects challenging. More data on the effects of variants on gene expression in *trans* will be important to understand *trans*-regulatory variation, both from high throughput approaches [112–118], and focused dissection of individual hotspots.

## Materials and methods

### Strains, plasmids, primers and media

Experiments were performed in haploid *S. cerevisiae* strains derived from S288C (BY4741 (MATa, his3Δ1 leu2Δ0 met15Δ0 ura3Δ0), referred to as "BY" in the text) and RM-11a, (RM HO(BY) (MATa, his3Δ1::CloNAT, leu2Δ0, ura3Δ0 HO(BY) allele) AMN1(BY allele), referred to as "RM"). All strains used in this study can be found in S8 Table. The HO(BY) allele was

introduced into this RM strain by replacing the hphMX cassette at HO with the BY allele in
YLK2442 (a gift from L. Kruglyak) by CRISPR-Swap. Importantly, the CloNAT resistance
gene at the *HIS3* locus in RM HO(BY) is not recognized by the gCASS5a. All plasmids used in
this study are in S9 Table. All primers/oligonucleotides are in S10 Table.

We used the following media (recipes are for 1L):

YNB+2% Glu +all (6.7 g yeast nitrogen base with ammonium sulfate and without amino
acids, 20 g glucose, 50 mg histidine, 100 mg leucine, 50 mg methionine, 200 mg uracil and ster-
ilized by filtration)

YNB+2% Glu -Leu (YNB+2% Glu+all (without leucine)

YPD (10 g yeast extract, 20 g peptone, 20 g glucose)

SDC-Leu (1.66 g SC -His -Leu -Ura, 50 mg histidine, 200 mg uracil, 20 g glucose)

SDC-His (1.66 g SC -His -Leu -Ura, 100 mg leucine, 200 mg uracil, 20 g glucose)

LB (10 g tryptone, 10 g NaCl, 5 g yeast extract)

Media for selection of resistance gene expression was supplemented at the following con-
centrations: ampicillin (100 μg/ml), nourseothricin sulfate (100 μg/ml), G418 sulfate (200 μg/
ml), hygromycin B (300 μg/ml). For solid media, 20 g/L agar was added prior to autoclaving.
Yeast were grown at 30˚C. Bacteria were grown at 37˚C.

## Plasmid construction

To construct the CRISPR-Swap plasmids we annealed oligos OFA0185 and OFA0186 for
gCASS5a (pFA0055) and OFA0552 and OFA0553 for gGFP (pFA0057), and ligated them into
the BclI and SwaI sites of pML107, a gift of John Wyrick (Addgene #67639) as described in
[45].

We have deposited pFA0055 and pFA0057 at Addgene under #131774 and #131784,
respectively.

The yeVenus reporter plasmids were created by PCR fusion of the *OLE1* (-1 to -936) or
*SDS23* (-1 to -1090) promoter fragment with the open reading frame of yeVenus, a gift of Kurt
Thorn (Addgene plasmid #8714) [119]. To create plasmids pRS415-p*OLE1*(BY)-yeVenus and
pRS415-p*OLE1*(RM)-yeVenus, the *OLE1* promoter and yeVenus PCR fragments were digested
with HindIII and BglII and ligated into pRS415 [120] digested with HindIII and BamHI. To
create pRS415-pOLE1(BY-FAR-RM)-yeVenus and pRS415-pOLE1(RM-FAR-BY)-yeVenus
the *OLE1* promoter PCR fragment was digested with HindIII and PstI and ligated into the
same sites of pRS415-OLE1(BY)-yeVenus_1. To create pRS415-SDS23(BY)pVenus, the *SDS23*
promoter and yeVenus PCR fragment was digested with SalI and NdeI and cloned into the
same sites of pRS415-pOLE1(BY)-yeVenus_1. For details on the creation of the PCR fusions
see S11 Table.

pRS415-OLE1(BY) was created by ligating the OLE1(BY) gene (-936 to+373) after PCR
amplification and digestion with HindIII (native in OLE1) and BamHI (on OFA0641 primer)
into the same sites of pRS415. pRS415-SDS23(BY) was created by ligating the SDS23(BY) gene
(-1033, +4730) after PCR amplification and digestion with BamHI and SacI into the same sites
of pRS415. pRS415-SDS23-OLE1p(BY) was created by ligating the 1009-bp intergenic region
between SDS23 and OLE1 after PCR amplification and digestion with BamHI and HindIII
into the same sites of pRS415.

## Tagging genes and cassette insertion

Insertions of cassettes for genome modification were performed using a standard PCR-based
one-step method [50]. Selection markers used were HIS3MX6, which allows growth of *his3*-
mutants without exogenous histidine, KanMX4, which allows growth with G418, natMX6,

which allows growth with nourseothricin sulfate/CloNAT and hphMX4 or hphNT1, which allows growth with hygromycin B. For C-terminal tagging of *HXT1*, we used the GFP-HIS3MX6 cassette for tagging of *HXT1*, and the mCherry-hphNT1 cassette, a gift of Jiří Hašek (Addgene plasmid #74635) [121] for tagging of *OLE1*. For the first step of CRISPR-S-wap, we used KanMX4 and hphMX cassettes for gene deletion or for cassette insertion without deletion. After selection for these markers, the transformants were single colony purified and insertion of the cassette in the correct location and absence of the wild-type allele were verified by PCR.

## Construction of repair templates for CRISPR-Swap

Repair templates were PCR amplified from BY and RM genomic DNA with primers designed to create products with termini homologous (ranging from 84–338 bp) to the region flanking the targeted cassette and, when possible, to be free of BY/RM sequence variants. To create hybrid BY and RM repair templates, we used PCR SOEing techniques [122]. See S11 Table for details on construction of each template.

## CRISPR-Swap

Strains were transformed with the gCASS5a or gGFP plasmid and a PCR-generated repair template using a standard lithium acetate procedure [123]. For each transformation, we used 25 ml of cells at $OD_{600}$ = 0.4–0.8. To prepare 50 ml of cells for transformation, the cells were pelleted by centrifugation at 3,000 g for 3 min, the supernatant removed and the cells were resuspended in 1 ml water and transferred to a 1.7 ml microfuge tube. The cells were pelleted at 2,500 g for 2 min, the supernatant removed, and the cells were resuspended in 1 ml of Solution 1 (0.1M LiAc, 1X TE buffer). The cells were pelleted once again, supernatant removed, and then resuspended in 200 μl of Solution 1. For each transformation, ~125 μl of the cell mixture was transferred to a 1.7 ml microfuge tube containing 100 ng of the guide RNA plasmid, 1000 ng of PCR-generated repair template and 5 μl (10 μg/μl) of salmon sperm carrier DNA (Sigma #D7656) and the tube was incubated on a turning wheel at 30˚C for 30 min. After which, 700 μl of Solution 2 (0.1M LiAc, 1X TE and 40% PEG 3350) was added and the mixture was returned to the turning wheel and incubated at 30˚C for 30 min. Next, the mixture was incubated at 42˚C for 15 min and then 500 μl of water was added. To wash the cells prior to plating, the cells were pelleted at 2,500 g for 2 min, the supernatant removed, and the cells were resuspended in 1 ml of water. The cells were pelleted once again at 2,500 g for 2 min, the supernatant removed, and the cells resuspended in 200 μl of water, plated onto two SDC-Leu plates, and incubated at 30˚C for 3–4 days. The median number of colonies growing on SDC-Leu after single-cut CRISPR-Swap was 38 for BY and 3 for RM strains. After double-cut CRISPR-Swap there were ~50% fewer transformants in each background. The resultant leucine prototrophic colonies were single-colony purified on SDC-Leu plates and then assayed for loss of the selectable marker cassette by identifying strains that could no longer grow on YPD with G418, or, for *OLE1* allele exchanges, YPD with G418 and/or YPD with hygromycin. We did not cure the strains of the CRISPR-Swap plasmid with the exception of the strains used for RNA sequencing and the RM HO(BY) strain YFA0254. In our experience, the plasmid is rapidly lost and therefore was likely not present during phenotyping of the strains. To cure the strains of the plasmid, the strains were single-colony streaked and then replica plated to SC-Leu to identify leucine auxotrophic colonies. We preserved a minimum of three independently derived strains for each allele swap in glycerol (15% final concentration) stored at -80˚C.

## Verification of allele exchange

We performed colony PCR to verify the absence of the selectable marker cassette(s) and presence of the desired allele. Further verification of variant incorporation was in some cases performed by sequencing or by restriction enzyme digestion. To verify variants by enzyme digestion, we screened for the presence of an SspI site created by the *OAF1(L63S)*-BY allele and an NcoI site created by the *OLE1(L304M)*-RM allele. We did not always verify the allele exchange since we found that 100% of the colonies that no longer expressed the selectable marker had the desired allele. Special cases of allele exchange verification are described below.

The *RGT2* repair templates in some allele swaps contained a single indel variant within the 5' homologous flanking region. In these strains, the variant was sequenced and only strains with the desired variant were preserved when possible. In other cases, the mismatched variant is indicated in Fig 2 and S8 Table. We found this variant to have no effect on expression of Hxt1-GFP.

The BY *FAA4-GFP OLE1(L304M)* strains were created using single-cut CRISPR-Swap of the KanMX cassette in BY *FAA4-GFP DAmP(OLE1)* (YFA0547) and a repair template amplified from BY genomic DNA with OFA0519, which carries the RM variant at L304M, and OFA0120. Because the site of homologous recombination can initiate anywhere along the *OLE1* gene present in the genome, the L304M variant was verified by restriction digestion and sequencing and 4 of 10 allele swaps successfully incorporated the RM variant at L304M.

When performing double-cut CRISPR-Swap at the *OLE1* locus, we observed incorporation of unexpected BY/RM chimeric alleles in 2/36 strains in which we genotyped at least one variant. We believe this can occur because the *OLE1* repair template is homologous to the sequence between the two cassettes, allowing recombination to occur between the repair template and the intervening sequence. Thus, we recommend verifying all variants along the length of the exchanged allele after performing double-cut CRISPR-Swap to ensure complete incorporation of the desired allele.

## Phenotyping of engineered strains

Precultures were inoculated with cells from strains freshly growing on YPD plates or from glycerol stocks and grown overnight in 800–1000 μl of YNB+2% Glu+all medium in a 2-ml deep-96-well-plate on an Eppendorf Mixmate shaker set at 1100 rpm at 30°C. The precultures were diluted to an $OD_{600} = 0.05$ in 100 μl of the same medium in a 96-well flat bottom plate (Costar) and the plates were sealed with a Breathe Easy membrane (Diversified Biotech). Each strain was measured in multiple technical replicates (different wells of the same clone), and most strains were additionally measured as multiple biological replicates (different clones picked from the same transformation). These replicates were distributed across different sectors of the plates, to guard against potential edge effects. All plate layouts are available with the plate reader data (see link below). The strains were phenotyped in a BioTek Synergy H1 (BioTek Instruments) plate reader at 30°C with readings taken every 15 min for 97 cycles with 10 sec of orbital shaking between reads and 11–13 min between cycles. Cell growth was characterized using absorbance readings at 600 nm and protein expression was measured using fluorescence readings taken from the bottom of the plate with the following parameters: GFP (excitation 488, emission 520 nm) and yeVenus (excitation 502, emission 532 nm) with an average of 10 reads per well and gain set on extended; and mCherry (excitation 588 nm, emission 620 nm) with an average of 50 reads per well and gain set at 150.

All plate reader measurements are available at our code repository at https://github.com/frankwalbert/threeHotspots.

## Growth of *RGT2* strains in different glucose concentrations

*HXT1*-GFP tagged strains, BY (YFA0276-8), BY *RGT2(V539I)* (YFA0489-91), RM (YFA0279-81) and RM *RGT2(I539V)* (YFA0492-95) were precultured overnight in 1 ml of YNB+all with the indicated glucose concentration (1%-12% glucose w/vol.) The cultures were diluted in 100 μl of the same media and phenotyped as described above. We removed one strain (YFA0275, a BY wild type strain present on one plate) from the analyses because it showed highly unusual *HXT1*-GFP expression at high glucose.

## yeVenus reporter expression

Strains BY (YFA0993), RM (YFA0254), or BY *OAF1(L63S)* (YFA0907) were transformed with pRS415-based plasmids containing *OLE1 / SDS23* intergenic fragments driving expression of the yeVenus reporter. Phenotyping was performed as described above in YNB+2% Glu -Leu to maintain the plasmids. Precultures were inoculated with transformants that were single colony purified on SDC-Leu plates or using pools of transformants (~5–10 colonies) taken directly from the SDC-Leu transformation plates and grown overnight, diluted, and phenotyped as described above.

## Modulation of OLE1 expression using Z₃EV

The BY *FAA4-GFP* strain expressing the estradiol responsive transcription activator was created by inserting p*ACT1*-Z$_3$EV-NATMX [64] at the *HO* locus, S11 Table for more details. The *OLE1* gene was then modified to have the Z$_3$EV responsive promoter cassette (KANMX-pZ$_3$EV and a C-terminal fusion with mCherry-hphNT1. Cells of BY *FAA4-GFP OLE1-m-Cherry* (YFA1105), BY *FAA4-GFP OLE1(FAR-RM)-mCherry* (YFA1140) and BY *HO::pACT1_Z$_3$EV FAA4-GFP Z$_3$EVpOLE1-mCherry* (YFA1110; glycerol stock prepared from a culture grown in 8 nM estradiol) were precultured overnight in 250 μl of YNB+2% Glu+all with 4, 5, 6, 8, 10, 15, 20, 30 or 40 nM of estradiol (Sigma E1024: 10 μM stock solution in ethanol), diluted in 100 μl of the same media, and phenotyped as described above.

## Extraction and sequencing of mRNA

Strains BY(YFA1130), BY *RGT2* back to WT(YFA1131), BY *OAF1(L63S)* (YFA1132), BY *OLE1(FAR-RM)* (YFA1133) and BY *RGT2(V539I)* (YFA1134) were precultured (7 precultures of each strain) in 1 ml of YNB+2% Glu+all as described above. After ~18 hours of growth, the precultures were diluted to OD$_{600}$ = 0.05 in 1 ml of the same media. After ~7 hours of growth, the optical density of the cultures was measured in the plate reader and growth was continued until an average plate OD$_{600}$ = 0.37, at which time the plate was centrifuged for 3 min at 2,100 g, the supernatant removed, and each cell pellet was resuspended in 1 ml of H$_2$O and transferred to a 1.7 ml microfuge tube. The tubes were centrifuged for 2 min at 16,000g, the supernatant removed, and the cell pellets were immediately frozen by immersion of the tube in liquid nitrogen and stored at -80˚C.

Five cell pellets from each strain were chosen for RNA isolation so that the average OD$_{600}$ within and between strains were as uniform as possible, with an average OD$_{600}$ = 0.37 (range of 0.34 to 0.40). Total RNA was isolated from cell pellets in 5 batches (each batch contained one of each of the 5 strains) using the ZR Fungal/Bacterial RNA mini-prep kit including the DNase I on-column digestion step (Zymo Research). Each cell pellet was resuspended in lysis buffer and transferred to screw-capped tubes containing glass beads and the cells were broken open using a mini-beadbeater (Biospec Products) in 5 cycles of 2 min bead beating followed by 2 min at -80˚C. The total RNA was eluted in 50 μl of DNAse/RNAse free water and the RNA

Integrity and concentration was measured using an Agilent 2200 Tape Station. RINe scores ranged from 9.8–10, with an average RNA concentration of 137 ng/μl.

Poly-A RNA was extracted from 550 ng of total RNA using NEBNext Poly(A) mRNA Magnetic Isolation Module (NEB) and used as input into the NEB Ultra II Directional RNA library kit for Illumina (NEB E7760) in two batches. NEBNext Multiplex Oligos for Illumina (Dual Index Primers Set1) were used to amplify and barcode the libraries using the following cycling conditions: Initial Denaturation at 98°C for 30 s and 10 cycles of: 98°C for 10 s and 65°C for 75 s, followed by a 65°C extension for 5 min.

The amplified DNA was quantified using Qubit DNA HS. Sample concentrations ranged from 47–102 ng/μl. An equal concentration of each of the 25 barcoded libraries was pooled and the average fragment size of the library was 350 bp, as determined using an Agilent High Sensitivity chip. High-output sequencing of 76-bp single-end reads was performed on an Illumina NextSeq 550 at the University of Minnesota Genomics Core. An average of 14 million reads were obtained for each sample.

Sequencing reads are available at GEO as series GSE134169.

## Lipid measurements

Strains BY (YFA0897), RM HO(BY) (YFA0254), BY *OAF1(L63S)* (YFA0907), BY *OLE1 (FAR-RM)* (YFA0914) and BY *OLE1(FAR-RM)OAF1(L63S)* (YFA1096) were precultured (6 precultures of each strain) for ~18 hours in 1 ml of YNB+2% Glu+all and then diluted in 7 ml of the same media to an $OD_{600}$ = 0.002 in a loosely capped 16 X 150 mm culture tube and grown at 30°C on a turning wheel until an approximate $OD_{600}$ = 0.34 (measured in plate reader). Samples were centrifuged for 3 min at 2,100g, the supernatant was removed and the cell pellets were resuspended in 1 ml of water and transferred to 1.7 ml microfuge tubes. The cells were pelleted for 2 min at 16,000g, the supernatant removed, and the cell pellets were immediately frozen by immersion of the tube in liquid nitrogen and stored at -80°C.

Cell pellets were shipped on dry ice to The Mayo Clinic Metabolomics Core (Rochester, MN) for analysis. Five replicate cell pellets from each strain were resuspended in 1X PBS and sonicated. Fatty acids and total lipid composition were measured against a standard curve on a triple quadrupole mass spectrometer coupled with an Ultra Pressure Liquid Chromatography system (LC/MS) as previously described [124]. Briefly, the cell pellets were spiked with internal standard prior to extraction with tert-Butyl Methyl Ether (MTBE). Roughly 25% of the sample was dried down, hydrolyzed, re-extracted and brought up in running buffer for the analysis of total fatty acid composition. The remaining portion of the extract was dried down and brought up in running buffer prior to injecting on the LC/MS for the NEFA measurement. Data acquisition was performed under negative electrospray ionization condition.

## Allele frequencies of the variants in the population

The phylogenetic tree used to describe the evolution of the causal variants was obtained from [67] (personal communication). Briefly, the tree was formed from the analysis of 1,544,489 biallelic sites across 1,011 *S. cerevisiae* strains using the R package "ape" [125] with the 'unrooted' method to display the tree. For each of the three variants, the matrix of variants from [67] was used to define which of the 1,011 strains carry the RM allele, the BY allele, both the BY and RM allele, or another allele. The edges of the tree were colored based on the alleles each strain carries. The absence of color continuity as seen with *OLE1* and *RGT2* can indicate multiple independent mutation events, but is more likely to arise from out-crossing events leading to mosaic genomes [126].

Alignments were performed using Clustal Omega (www.ebi.ac.uk/Tools/msa/clustalo/) with default settings. Sequences for alignment were retrieved from the NCBI nucleotide

database. Predictions of variant effect on the function of the Rgt2, Oaf1, Sds23, and Ole1 proteins were calculated using Provean (Protein Variation Effect Analyzer) (http://provean.jcvi.org).

## Computational and statistical analyses

All analyses described below were conducted in R (www.r-project.org), with individual packages indicated throughout. Figures were generated using base R and ggplot2 [127]. Analysis code is available at https://github.com/frankwalbert/threeHotspots.

## Quantification of gene expression from plate reader data

We used the 'growthcurver' R package [128] to fit a logistic growth curve to the $OD_{600}$ readings in each well. Every plate had several blank wells that contained medium but no yeast, and we used these blank wells to correct for the optical density of the medium. We visually confirmed successful fit of the growth curve for every well, and additionally excluded any wells for which growthcurver indicated poor model fit. From the fitted growth model, we extracted growth rates as well as the "inflection point" of the growth curve, i.e. the time point at which the population reached half its maximum capacity. We chose this time point for our measure of expression because in practice it closely matches the $OD_{600}$ values used to map the hotspots [12], and because while cultures are still growing exponentially at this time point, they have reached a high enough density to allow accurate quantification of fluorescence.

To obtain expression values for downstream analyses, we subtracted the mean $OD_{600}$ and mean fluorescence of the blank wells included on each plate from all other wells on the plate. We calculated the mean of three background-subtracted time points centered on the inflection point (Fig 1C) for $OD_{600}$ and fluorescence. Downstream analyses of gene expression used the ratio of this mean fluorescence divided by the mean $OD_{600}$.

Prior to statistical analysis, we $log_2$-transformed these fluorescence ratios. Every plate that carried strains with the BY and RM backgrounds also carried untagged wild type control strains without any fluorescent markers. We subtracted the average $log_2$(fluorescence ratio) values from these untagged strains from those of the tagged strains. Thus, the fluorescent phenotypes used in statistical analyses and displayed in the figures are in units of $log_2$-fold change compared to untagged strains with matched genetic background.

## Statistical analyses of plate reader data

For fine-mapping, we used pairwise comparisons of genotypes to determine whether the expression associated with a given edited allele differed significantly from the wild type or other alleles. These pairwise tests were computed using mixed linear models whose random effect terms depended on the structure of the data available for each comparison. Specifically, two random terms were included where appropriate:

**Plate identity.** As fine-mapping progressed, most genotypes were measured on several plate runs, usually along with different sets of other genotypes. This resulted in a complex data structure in which genotypes were sometimes but not always included on the same plates, and were run over a span of several months. To account for this structure, the model included plate identity as a random term. To ensure that plate effects were properly accounted for in both genotypes in a given pairwise comparison, each comparison was computed using only data from plates that carried both genotypes under consideration. For example, while wildtype strains were run on multiple plates, a given edited strain may only have been present on one of these plates. In this scenario, only this one plate would be used in the statistical comparison. Note that data in figures in the paper that display plate reader data from multiple plates were

not corrected for plate effects. We chose to not correct for plate effects in the plots because we wished to present a raw view of the data, and because there is no good way to apply a common visual correction for plate identity when different genotype comparisons require different plate corrections, depending on which genotypes were present on each plate. In the figures, plate identity is indicated with different symbols (dots, squares, triangles, etc.). Position of the well within the plate (e.g. edge vs center) was not considered in our models.

**Clone identity.** During strain engineering, we created at least three independent clones of each strain. Clone identity was included in the model as a random effect to control for any systematic differences among these clones. In the figures, we visually grouped data from different wells for a clone by connecting these wells with a line. For the yeVenus reporter experiments, we collected data from individual transformed colonies as well as from small transformant pools that each contained (and effectively averaged) multiple colonies. For statistical analyses, we treated each small transformant pool as if it were a colony.

In the equations below, we denote random effects in parentheses. For each genotype, we used the "lmer" function in the lme4 package [129] to fit a model of the phenotype *y* with the above random effect terms as appropriate:

*H0*: *y = (plate) + (clone) + ε*

where *ε* is the residual error. We fit a second model that includes a fixed effect term for genotype identity:

*H1*: *y = (plate) + (clone) + genotype + ε*

We tested for significance of the genotype term using ANOVA comparing *H0* and *H1*. Note that for a few comparisons in which only a single clone was run in replicate on a single plate, we did not include random effects terms. We fit these models using the "lm" function.

P-values in plate reader analyses during fine mapping were not corrected for multiple testing.

## Tests for non-additive genetic interactions

To test for non-additive interactions of a given allele (i.e., the BY or RM allele at a given causal variant) with the strain background (BY or RM), we fit a model with a fixed effect term for strain background, in addition to a term for the allele at the variant of interest:

*H0*: *y = (plate) + (clone) + strain + allele + ε*

and a second model that adds an interaction term between allele and strain background:

*H1*: *y = (plate) + (clone) + strain + allele + strain:allele + ε*

We used ANOVA to test if the inclusion of the *strain:allele* interaction term in *H1* significantly improved model fit. These interaction tests only considered plates that contained both alleles for the given variant in both strain backgrounds. We used the same models to test for interactions between FAR alleles and *OAF1* alleles by replacing the "RM" factor level in *strain* with "BY (*OAF1*-RM)".

To compare the *OAF1*/*OLE1* allele interaction to that observed earlier in eQTL data (Source Data 14 in [12]), we obtained expression values (Source Data 1 from [12]) and used a linear model to regress out effects of collection batch and OD$_{600}$ for these data (Source Data 2 from [12]). We then used genotypes (Source Data 3 from [12]) at the marker positions corresponding to the two interacting loci at *OAF1* and *OLE1* to divide the segregants in [12] into four two-locus genotype classes and plot their *FAA4* mRNA levels in Fig 7A.

## Dependence of the *RGT2(V539I)* effect on glucose concentration

To test if the effect of the causal variant in *RGT2* depended on glucose concentration in the medium, we extended our modeling framework to include *glucose* (log$_2$(glucose concentration))

as a numeric covariate. We fit a model that included all possible pairwise interactions between strain, allele, and glucose:

H1: y = (plate) + (clone) + strain + allele + glucose + strain:allele + strain:glucose + allele:glucose + ε

and compared this model to simpler models from which we dropped the respective interaction term of interest. For example, to test for the interaction between glucose and the allele effect:

H0: y = (plate) + (clone) + strain + allele + glucose + strain:allele + strain:glucose + ε

We computed p-values comparing these models using ANOVA.

## RNASeq data handling

We used trimmomatic [130] version 0.38 to trim Illumina adapters, filter out reads shorter than 36 bp, trim bases with a quality score of less than 3 from the start and end of each molecule, and perform sliding window trimming to remove bases with an average quality of less than 15 in a window of four bases. This filtering retained ≥97% of reads. We used kallisto [131] to pseudoalign these trimmed and filtered reads to the *S. cerevisiae* transcriptome obtained from Ensembl [132] build 93 based on genome version sacCer3 [133]. Following recommendations in [134], we used FastQC [135] and RSeQC [136] to examine the quality of our 25 samples and found them to all be of high quality and, importantly, highly similar to each other. We retained all 25 samples for downstream analysis.

As a measure of gene expression, we used "estimated counts" in the kallisto output for each gene in each sample. To exclude genes with poor alignment characteristics, we used RSeQC to calculate Transcript Integrity Numbers (TINs) per gene and sample, and also considered "effective gene length" produced by kallisto. We retained genes in which no sample had a count of zero, no sample had a TIN of zero, and with effective length larger than zero. This filtering retained 5,400 genes (out of 6,713 annotated) for further analysis.

## RNASeq statistical analyses

Statistical analyses were conducted using the DESeq2 R package [137]. During RNA isolation and sequencing library generation, we had collected a number of covariates and batch identities: Bioanalyzer-based RNA Integrity Number (RIN), Bioanalyzer-based RNA concentration, Qubit-based RNA concentration, $OD_{600}$ of the culture at time of flash freezing, as well as batch for cell harvest, RNA isolation, and library generation. Samples from our 5 genotypes had been distributed equally among these three batches. We examined the influence of these technical covariates by comparing them to principal components computed on variance-stabilized data [137] and found that the three batches (in particular cell harvest) appeared to influence the results. We thus included these three batches as covariates in all further analyses. We used surrogate variable analysis (SVA) [138] to further account for unexplained technical variation and included two SVs in our statistical model. While choices about which specific technical covariates and SVs to include in the model did slightly alter the significance tests for individual genes, our main result of positive correlations between hotspot effects and differential expression was robust to these choices.

We fit the DESeq2 model to all 25 samples and conducted pairwise tests for differential expression between genotypes. Specifically, we compared the edited *OAF1* and *OLE1* alleles to a BY wildtype strain (YFA1130) engineered by CRISPR-Swap with gGFP to remove the GFP tag from *FAA4*. We compared the edited *RGT2* variant to a BY wild type strain (YFA1131) engineered by CRISPR-Swap with gCASS5a to replace rgt2Δ::kanMX6 with the *RGT2* BY allele. The main text describes differential expression results based on either nominal p-values (p < 0.05), or based on a multiple-testing corrected threshold computed via false discovery rate estimation using the Benjamini-Hochberg method [139] as reported by DESeq2.

To compare differential expression to hotspot effects, we used $log_2$-fold changes estimated by DESeq2 and hotspot effects that had been estimated by fitting a lasso model to all expressed genes and all 102 hotspots, as described in [12]. These hotspot effects were obtained from Source Data 9 in [12]. We used nonparametric Spearman rank correlation to compare differential expression with the hotspot lasso coefficients. We excluded genes with a hotspot effect of zero. While inclusion of these genes slightly degraded the magnitude of the correlations between hotspot effects and differential expression, all correlations remained significant and positive. We computed two-sided Fisher's exact tests to test if genes with significant differential expression are more likely than expected to have nonzero hotspot effects.

Gene ontology enrichment analysis for genes with strong hotspot effects but no differential expression was conducted using the "Gene Ontology Term Finder" tool on SGD [140]. As the background set, we used all genes present in both the hotspot effect matrix and our RNASeq data. As the test set, we used genes with an absolute hotspot effect of at least 0.3 and a differential expression p-value larger than 0.3.

## Statistical analyses of lipid data

To correct for possible technical confounders from lipid composition and NEFAs, we used a linear model to regress out effects of acquisition order and sample grouping. For NEFAs, we also regressed out total protein, which had been measured from the same samples. The residuals from these regression were used in the statistical analyses below. To obtain total saturated, unsaturated, C18, and C16 measures, we summed the measures for the respective individual lipid species in these groups.

To analyze the effects of the *OAF1* and *OLE1* alleles in the BY background, we jointly considered the measures from the four genotypes (BY, BY(*OAF1-L63S*), BY(FAR-RM), and BY with both RM alleles) by fitting a linear model to each lipid compound *y* that models the effects of the *OAF1* and the FAR allele at *OLE1*:

$$y = OAF1 + OLE1 + OAF1{:}OLE1 + \varepsilon$$

where $\varepsilon$ is the residual error. We analyzed this model using ANOVA to test for main effects of each allele as well as for the interaction term. P-values were not corrected for multiple testing.

The BY and RM backgrounds we compared using T-tests.

## Supporting information

**S1 Fig. Guide RNA recognition sequences.** A. Schematic of a cassette typically used for gene deletions. The gCASS5a recognition sequence is marked with a bracket and the PAM site is underlined. Start of the TEF promoter sequence driving expression of the selectable marker is in red letters. B. Schematic of a cassette used for C-terminally tagging open reading frames with GFP. The location of the gGFP recognition sequence is marked with a bracket and the PAM site is underlined. The start of the GFP sequence is in neon green. The recognition sites for SalI (pink) and PacI (purple) and the Cas9 cleavage sites (scissors) are shown to allow easy comparison of the gRNA recognition sequences, which are specific to each cassette. Designated with arrows are the universal primer sequences, S1 or U2 and F1, used for amplification of common cassettes. (TIFF)

**S2 Fig. *OAF1* fine mapping.** On the left are schematics of *OAF1* alleles with BY sequences in blue and RM sequences in red. Missense variants are marked with a straight line. For clarity, synonymous and non-coding variants are not depicted. On the right are the corresponding Faa4-GFP fluorescence levels for each allele. P-values are for tests comparing each allele to its respective wildtype. Significant p-values are outlined. Blue boxplots indicate alleles in the BY

background and red boxplots and background gray shading indicate alleles in the RM background. Lines group measurements of the same clone. Different symbols (circles, squares, etc.) denote different plate reader runs.
(TIFF)

**S3 Fig. *OLE1* fine mapping.** On the left are schematics of *OLE1* alleles with BY sequences in blue and RM sequences in red. Only the one missense variant and none of the synonymous variants in the open reading frame are marked. Variants in the non-coding region are maked with a single line for a SNV and a two diagonal lines for INDELs. On the right are the corresponding Faa4-GFP fluorescence levels for each allele. P-values are for tests comparing each allele to its respective wildtype. Significant p-values are outlined. Blue boxplots indicate alleles in the BY background and red boxplots and background gray shading indicate alleles in the RM background. Lines group measurements of the same clone. Different symbols (circles, squares, etc.) denote different plate reader runs.
(TIFF)

**S4 Fig. yeVenus reporter expression.** On the top are schematics of the SDS23/OLE1 locus and the two orientations of the yeVenus reporter constructs. The bottom panel shows yeVenus fluorescence levels for the indicated yeVenus reporter constructs. Blue boxplots indicate alleles in the BY background and red boxplots indicate alleles in the RM background. Lines group measurements of the same clone. Different symbols (circles, squares, etc.) denote different plate reader runs.
(TIFF)

**S5 Fig. Effects of plasmid overexpression of SDS23/OLE1 sequences on Faa4-GFP expression.** Faa4-GFP fluorescence levels of strains transformed with a *LEU2*-CEN plasmid containing the indicated sequence. Lines group measurements of the same clone.
(TIFF)

**S6 Fig. Growth rates as a function of estradiol dose.** Error bars show standard deviations.
(TIFF)

**S7 Fig. Lipid and fatty acid measurements.** All individual measurements are shown. For each genotype, the figure shows values for each replicate (smaller points) along with the mean (larger points) and standard deviation (vertical lines).
(TIFF)

**S1 Table. Fine-mapping p-values.**
(XLSX)

**S2 Table. Lipid measurements.**
(XLSX)

**S3 Table. Lipid statistics.**
(XLSX)

**S4 Table. RNA sequencing gene expression counts.**
(XLSX)

**S5 Table. RNA sequencing sample and batch information.**
(XLSX)

**S6 Table. RNA sequencing results.**
(XLSX)

**S7 Table. eQTL hotspot literature review.**
(XLSX)

**S8 Table. Yeast strains.**
(XLSX)

**S9 Table. Plasmids.**
(XLSX)

**S10 Table. Oligos.**
(XLSX)

**S11 Table. Yeast strain and plasmid construction.**
(XLSX)

## Acknowledgments

We thank Mahlon Collins, Randi Avery, Krisna Van Dyke, Joshua Bloom, and Meru Sadhu for critical reading and comments on the manuscript. We thank Leonid Kruglyak, David Kirkpatrick, Melissa Gardner, and Scott McIsaac for yeast strains and plasmids, and Anne Friedrich for help with the 1,011 yeast isolate genome data.

## Author Contributions

**Conceptualization:** Sheila Lutz, Christian Brion, Frank W. Albert.

**Data curation:** Sheila Lutz, Frank W. Albert.

**Formal analysis:** Sheila Lutz, Christian Brion, Margaret Kliebhan, Frank W. Albert.

**Funding acquisition:** Frank W. Albert.

**Investigation:** Sheila Lutz, Frank W. Albert.

**Methodology:** Sheila Lutz, Frank W. Albert.

**Project administration:** Frank W. Albert.

**Software:** Margaret Kliebhan, Frank W. Albert.

**Supervision:** Frank W. Albert.

**Validation:** Sheila Lutz.

**Visualization:** Sheila Lutz, Christian Brion, Frank W. Albert.

**Writing – original draft:** Sheila Lutz, Frank W. Albert.

**Writing – review & editing:** Sheila Lutz, Christian Brion, Frank W. Albert.

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
