## [Decision Letter · Decision Letter 0]

1 Oct 2019

Dear Dr Albert,

Thank you very much for submitting your Research Article entitled 'DNA variants affecting the expression of numerous genes in trans have diverse mechanisms of action and evolutionary histories' to PLOS Genetics. Your manuscript was fully evaluated at the editorial level and by independent peer reviewers. The reviewers appreciated the attention to an important topic but identified some aspects of the manuscript that could be improved.

The reviewers were all very impressed with the quality of work. The comments are thoughtful and will improve the quality of the manuscript. However, none of the suggested changes are essential. Thus, consider this as a manuscript that will be accepted, even though it is in the system as a minor revision to allow you to incorporate the edits and other comments into the manuscript.

We therefore ask you to modify the manuscript according to the review recommendations before we can consider your manuscript for acceptance. Your revisions should address the specific points made by each reviewer.

[LINK]

Yours sincerely,

Justin C. Fay

Associate Editor

PLOS Genetics

Hua Tang

Section Editor: Natural Variation

PLOS Genetics

Reviewer's Responses to Questions

**Comments to the Authors:**

Reviewer #1: In the manuscript, “DNA variants affecting the expression of numerous genes in trans have diverse mechanisms of action and evolutionary histories”, the authors provide a thorough and interesting investigation of three trans eQTLs implicated as hotspots for expression change, and in the process present a technique for allele swaps that will be of broad interest to the yeast community. The manuscript is written clearly and compellingly, and addresses a broad and important question. It ranges from detailed functional analysis to an investigation of the phylogenetic distribution of variants. As I read the manuscript, every technical and statistical question that arose ended up being addressed. In fact, I commend the authors on the detailed methods, the rationale for the technical and statistical choices, and the inclusion of all the results and data. With regard to the journal’s criteria: originality, broad interest, solid methodology, and support for conclusions, the manuscript hits all those points. As such, I only have general comments and specific minor suggestions.

First, with regard to the main question of manuscript, which is determining the molecular basis of trans eQTLs, there is no a priori reason to assume that one type of molecular mechanism will be more common than another (i.e., I don’t think it is a surprise that trans-acting eQTLs can be conserved or not, regulatory vs. coding, functionally diverse proteins, etc.). In fact, given the delicate dance that goes on within cells that sense and respond to their internal and external environments, it is not surprising that all mechanisms can be found. Using descriptors like “remarkable diversity” seems a little too much. That being said, it is important to demonstrate that all types of mechanisms are possible with solid examples and this study does that incredibly well.

Second, I think that it is quite interesting that all of the derived alleles seem to be in the direction of null alleles, given the current public conversation about loss-of-function mutations (https://onlinelibrary.wiley.com/doi/full/10.1111/evo.13710). While it may out of the scope of this manuscript, a sentence or two in the discussion discussing the relevance of these findings would put this observation in a broader context.

Third, I strongly suggest the authors give a brief description of the study that this paper follows up on. For example: How many eQTLs were found in this study (what was the punchline)? What media was used (this matters later on when glucose level in the medium is tested)? I understand the three loci chosen from that study for dissection were chosen because they were mapped to a small genomic region (which makes sense), but a little bit of an overview of the previous work would be helpful.

Finally, with regard to the RGT2 story, how do the levels of glucose tested compare to anything the yeast might see in the environment? Are they comparable? Is it possible to test 0.5% and 0.1%? What level was in the medium of the original study in which this locus was implicated? Since the paper goes on to look at the distribution of these alleles, it is worth thinking about why we see the distribution we do and what the strains in the phylogenetic tree are likely to experience.

Minor:

It is a bit hard to see the shapes of the points in Figure 5C. There also seems to be more variation in these data than the others. Can the authors comment on why they think that is in the manuscript (I suspect it is simply because the effect of the SNV is smaller).

In figure 4C, it took a bit for me to comprehend the results. This is in part because in figure 2, the allele swaps essentially recapitulate the two different wildtype levels and that is what the reader expects to see (the comparable figures for the other alleles are in the Supplemental Material). However, in figure 4, swapping the causal variant does not lead to the wt level of the other background. The authors might consider putting small arrows next to the RM and BY labels on the x-axis where the allele is specified. A little up arrow next to BY and down arrow next to RM would remind the reader of the expected direction of the effect of the allele swap. But this is a style choice for the authors.

Line 172: “There” should be “The”

Reviewer #2: This manuscript describes fine mapping of previously identified eQTL hotspots down to the causative polymorphisms. The authors use a two-step “CRISPR-SWAP” strategy that can be adapted to swap essential genes. The authors validate the approach on three candidate hotspot genes, using GFP fusions to genes affected by each hotspot. The causative variants included two missense variations and one cis variant that affected a DNA-binding site. The authors explored both GxE and epistatic interactions across hotspots and found evidence for both. The authors performed RNA-seq on causative alleles to show that they affected more than just the single reporters tested. Beyond gene expression, the authors show that two hotspots that affect genes involved in fatty acid metabolism indeed have allele-specific effects on cellular lipid profiles. Overall, this paper is well written, the data is generally well presented, and the interpretations are mostly sound. I do offer some critiques for the authors to address. (And while I have quite a few comments, I should stress that these are mostly asking to clarify for the reader).

1) While I appreciate the advantages of the authors’ CRISPR-SWAP method, I think it’s fair to point out that CRISPR has been used before for allele swaps in yeast (PMID 29114020)

2) The authors claim that CRISPR-SWAP is 2-step, but is that true for the essential genes? Were the Kan and Hyg markers introduced simultaneously into the OLE1 region?

3) The authors claim that they obtained unexpected chimeric alleles for the OLE1 swap, which they interpret as being due to homology between the OLE1 repair template and the intervening sequence between the cassettes. This should be true of any swap of essential regions, right? I think a recommendation of sequencing of allele swaps for essential regions is warranted.

4) Line 1413: “and” instead of “are,” unless I am not understanding.

5) For the completeness of the literature review of hotspots, Oaf1 was at least hypothesized to be causative for an eQTL hotpsot in PMID 19223586.

6) Line 348: Instead of calling the CEN plasmids single copy, it is probably better to call them low copy (see PMID 23107142).

7) I’m not sure what to make of for SDS23 overexpression having an effect in Fig S5. At first I thought that maybe having extra copies of the FAR sequence could affect TF binding at other sites, but the intergenic sequence doesn’t have that effect. Based on SDS23 function, do the authors have any thoughts?

8) Fig6: what do the lines above and below each point denote (e.g. standard deviation)?

9) For Figure S6, what does the y-axis denote? Growth rate is usually in the units time^-1, but negative values would not make sense then.

10) Line 368: The logic of why estradiol-inducible OLE1 requires higher expression than OLE1 under the native promoter is not clear to me. It seems far-fetched that the inability to tune OLE1 at lower expression levels would have such a dramatic effect on growth, while expressing much higher levels is apparently fine. Not that I can think of a better explanation. Note: this assumes that the effect really is dramatic given my inability to grok the y-axis for Fig S6.

11) For Figure 7A, I’m not sure that the blue/red color scale works. For the other figures, blue/red denotes BY vs. RM genetic background for allele swaps, which is what I thought this denoted at first. Of course, this makes no sense when looking at segregants for the eQTL mapping, which left me a bit confused until I noticed in the key that the colors denote OAF1 allele. It may just be me, but I think that other readers may get acclimated to the blue/red BY/RM scale.

12) While perhaps not statistically significant, variation in OLE1 FAR does seem to affect lipid composition with a similar direction for both the RM and BY OAF1 variants. Based on the directionality, this seems important enough to at least mention.

13) Line 433: The logic here does not make sense to me. How can reduced substrate consumption not also result in reduced product? I think there may be something going on at the total lipid level to compensate. Instead of just the NEFAs, what happens if you show the total sat/unsat for the total lipids?

14) For overlap of the hotspot genes with the DE transcripts for each variant, it would be useful to also look at the percent overlap of differentially expressed genes with the hotspot-affected genes and Fisher’s exact test for significance.

15) For Figure 9, it’s not clear to me what hotspot effect means. Is this essentially the difference in log2 expression when you sort on the segregants at that genotype? The methods of this paper point to the Albert et al. eLife paper, but I couldn’t really parse what “hotspot effect” meant from that paper.

16) For the Discussion, I would have liked to see more on the role of both Oaf1 and Ole1 on regulating FAA4 expression. Based on the function of Oaf1 in fatty acid metabolism, does it make sense that FAA4 expression would increase as Oaf1 activity decreases? Is there an Oaf1 site in the FAA4 promoter? Likewise for reduced Ole1 activity—would you expect compensatory FAA4 expression? Also, does FAA4 have a FAR site (and is it known what TF binds to that site)?

17) Lines 569-570: yeast are rather gene dense at the genome level, so the predominance of coding to non-coding variation may be a function of target size.

18) Line 574-576: while I think it’s most likely that enzymes exert hotspot effects via alterations of metabolites, I would also bet that they are more likely to have moonlighting functions than expected by chance. In the case of Ole1, I think the jury is out until mechanism is elucidated.

19) Lines 580-582: the authors speculate that the OLE1 FAR allele could exert its hotspot effects via Mga2. Are genes affected by the OLE1 hotspot more likely to Mga2 targets (by ChIP or expression data)?

Reviewer #3: This manuscript describes fine-mapping of the causal nucleotides for expression differences between Saccharomyces cerevisiae isolates BY and RM located in previously defined eQTL “hot spots” for trans expression variation. To do so, the authors introduce a CRISPR-Cas9 based method, “CRISPR-Swap”, analogous to traditional marker-assisted homologous recombination but using CRISPR-directed cutting against an integrated selection marker to generate a double-strand break that can then be repaired with a library of sequence variants. For each of three eQTL hot spots, the experiment reported here maps a causal variant affecting one focal gene linked to the eQTL in trans using a fluorescent reporter for the focal gene. Identifying the frequency of those causal variants in context of S. cerevisiae population variation is a nice additional flourish.

The problem of understanding trans contributions to gene expression variation is a hard one. Because the problem is so hard, there are some features of this manuscript that are unsatisfying, despite what I believe is ultimately a well-rounded and useful contribution to describing the genetic basis of expression variation. The authors tackle the problem of defining the causal nucleotides for gene expression differences within three eQTL hot spots, but analyze the consequences of sequence differences on a single focal trans gene in each case. (RNA sequencing after flipping of the fine-mapped causal nucleotide provides support for the conclusion that the fine-mapped variants alter expression for more than just the mapped gene, but with limited power.) Further, fine-mapping is performed across a subset of SNPs within the eQTL interval based on their overlap with candidate genes. This approach provides useful examples of how single nucleotides can influence trans expression differences. It cannot fully address whether the single variant explains all the trans effects mapped to eQTL hotspots or whether multiple SNPs colocalized in the same region might contribute to trans effects on different genes. Other groups have published methods that might provide the necessary firepower to do a higher throughput analysis of more variants to answer these questions (in some cases working on the exact same genetic cross), but applying those methods to comprehensively fine-mapping trans expression differences remains a non-trivial undertaking. This manuscript fills that gap by expanding the number of documented SNPs impacting expression in trans, and does so with a nice attention to the potential functional consequences, environmental dependence, genetic interactions, and population context of the SNPs mapped. The range of differences between even these three examples offer useful hypotheses moving forward for how segregating genetic variation can have cascading and interacting impacts on gene expression and cell physiology. As a result, the work provides a useful illustration of the complexity of understanding trans effects on gene expression variation.

Questions to address:

1. In the general the manuscript is well written and easy to follow. In the introduction, it would be helpful if the authors could (briefly) make the case for what is gained by knowing the causal nucleotide within an eQTL. (An additional sentence or so on this point would do it, but given the history of critique of the QTN program, why do it here?)

2. Please explain the choice of these three loci for fine-mapping over other eQTL hot spots identified in prior work.

3. The GFP screen taking into account the growth curve of the strain is elegant. I miss a little more experimental detail about how this screen was executed, including technical information about how replicates were compared within vs between independent runs, whether position or edge effects across a plate were taken into account, etc.

4.Is RGT2 itself plastic to glucose level?

5.Minor concerns:

a. The blue and red fill for boxplots are challenging to distinguish in areas where the confidence intervals are tight. In places like Fig 3, I suggest adding labels (perhaps in a row at the top of the plot?) to distinguish alleles.

b. The symbols that represent different plate reader runs are also very challenging to make out at current sizing. I understand the desire to minimize the visual impact of this technical information, but if provided, the size of the symbol should be large enough to read.

c. Is there a reason for the choice to display hot spot 95% confidence intervals for RGT2 and 90% confidence intervals elsewhere?

**Have all data underlying the figures and results presented in the manuscript been provided?**

Reviewer #1: Yes

Reviewer #2: Yes

Reviewer #3: Yes

PLOS authors have the option to publish the peer review history of their article (what does this mean?). If published, this will include your full peer review and any attached files.

Reviewer #1: No

Reviewer #2: No

Reviewer #3: Yes: Andrea Hodgins-Davis

---

## [Editor Report · Decision Letter 1]

28 Oct 2019

Dear Dr Albert,

We are pleased to inform you that your manuscript entitled "DNA variants affecting the expression of numerous genes in trans have diverse mechanisms of action and evolutionary histories" has been editorially accepted for publication in PLOS Genetics. Congratulations!

Yours sincerely,

Justin C. Fay

Associate Editor

PLOS Genetics

Hua Tang

Section Editor: Natural Variation

PLOS Genetics

Comments from the reviewers (if applicable):

**Data Deposition**

http://datadryad.org/submit?journalID=pgenetics&manu=PGENETICS-D-19-01353R1

**Press Queries**

---

## [Editor Report · Acceptance letter]

11 Nov 2019

PGENETICS-D-19-01353R1 

DNA variants affecting the expression of numerous genes in *trans* have diverse mechanisms of action and evolutionary histories 

Dear Dr Albert, 

We are pleased to inform you that your manuscript entitled "DNA variants affecting the expression of numerous genes in *trans* have diverse mechanisms of action and evolutionary histories" has been formally accepted for publication in PLOS Genetics! Your manuscript is now with our production department and you will be notified of the publication date in due course.

With kind regards,

Matt Lyles

PLOS Genetics

On behalf of:
